# Exploring the Design Space of Diffusion Bridge Models

**Shaorong Zhang, Yuanbin Cheng, Greg Ver Steeg**
Unversity of California Riverside
{szhan311, ychen871, gregoryv}@ucr.edu

## Abstract

Diffusion bridge models and stochastic interpolants enable high-quality image-to-image (I2I) translation by creating paths between distributions in pixel space. However, recent diffusion bridge models excel in image translation but suffer from restricted design flexibility and complicated hyperparameter tuning, whereas Stochastic Interpolants offer greater flexibility but lack essential refinements. We show that these complementary strengths can be unified by interpreting all existing methods within a single SI-based framework. In this work, we unify and expand the space of bridge models by extending Stochastic Interpolants (SIs) with preconditioning, endpoint conditioning, and an optimized sampling algorithm. These enhancements expand the design space of diffusion bridge models, leading to state-of-the-art performance in both image quality and sampling efficiency across diverse I2I tasks. Furthermore, we identify and address a previously overlooked issue of low sample diversity under fixed conditions. We introduce a quantitative analysis for output diversity and demonstrate how we can modify the base distribution for further improvements. Code is available at https://github.com/szhan311/ECSI.

## 1 Introduction

Denoising Diffusion Models (DDMs) and flow matching create a stochastic process to transition Gaussian noise into a target distribution [33, 14, 34, 19]. Building upon this, diffusion bridge-based models (DBMs) have been developed to transport between two arbitrary distributions, $\pi_T$ and $\pi_0$, including I2SB [21], DSBM [39], DDBM [18], DBIM [42], Bridge Matching [28]. DBMs achieve superior image quality in I2I translation compared to DDMs [18, 21, 2], primarily because the distance between source and target image distributions is typically smaller than that between Gaussian and target distributions.

While DBMs like DDBM [39], DBIM [42], and I2SB [21] achieve state-of-the-art FID scores in image-to-image translation, they suffer from limited design flexibility, constrained bridge path formulations, and complex parameter tuning. In contrast, Stochastic Interpolants (SIs) [1, 2] offer a simpler and more flexible framework, but they have yet to integrate practical advances from recent diffusion bridge models, such as preconditioning. Besides, SIs require training two separate models, unlike the more efficient single-model setup in DDBM. Table 1 summarizes the key characteristics of these methods, highlighting that their complementary strengths had not yet been unified.

Another overlooked issue stemming from restrictive design choices in previous bridge models is the lack of diversity in outputs. While some image translation tasks are one-to-one, we find that in one-to-many translation tasks, like black and white edges to color images, previous methods produce limited variation in colors and textures. We refer to this as the *conditional diversity* problem and show that our approach leads to significant improvements.

39th Conference on Neural Information Processing Systems (NeurIPS 2025).

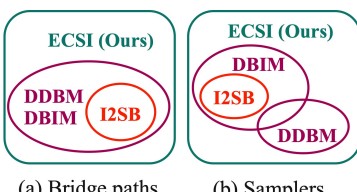

(a) Bridge paths  (b) Samplers

Figure 1: The design space of bridge paths and samplers.

|  | DDBM | DBIM | DSBM | SI | ECSI (ours) |
|---|---|---|---|---|---|
| Endpoint conditioning | ✓ | ✓ | ✗ | ✗ | ✓ |
| Uncoupled parameters | ✗ | ✗ | ✗ | ✓ | ✓ |
| Extensive bridge paths | ✗ | ✗ | ✓ | ✓ | ✓ |
| Extensive samplers | ✗ | ✗ | ✗ | ✓ | ✓ |
| Preconditioning | ✓ | ✓ | ✗ | ✗ | ✓ |
| Modified base density | ✗ | ✗ | ✗ | ✗ | ✓ |

Table 1: Characteristics of different bridge models.

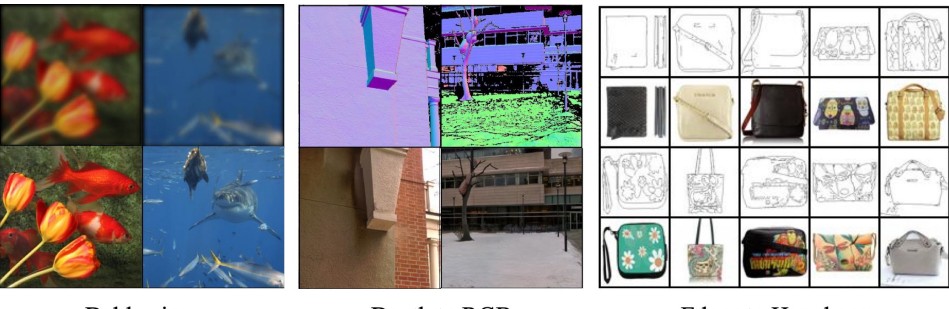

Deblurring  Depth to RGB  Edges to Hangbags

Figure 2: Samples for I2I translation with our ECSI models: Deblurring, Depth-RGB, and Edges to Handbags. For each pair of images, we show the input image (upper) and the output image (bottom).

Our main contributions are as follows:

- We propose **Endpoint-Conditioned Stochastic Interpolants** (ECSI), which extend stochastic interpolants by incorporating endpoint conditioning and preconditioning. Previous bridge methods artificially coupled unrelated aspects of the transition kernel. ECSI introduces a decoupled parametrization that expands and simplifies the design space for bridge paths and samplers. To further improve sampling quality and efficiency, we develop a novel noise control scheme and an efficient sampling algorithm.

- We identify a previously overlooked issue: the low diversity of outputs conditioned on fixed source images. To address this, we propose modifying the base distribution. Furthermore, to quantitatively evaluate conditional output diversity, we introduce a new metric—Average Feature Diversity (AFD).

- Experimental results demonstrate our model's state-of-the-art performance in both image quality and sampling speed across various I2I tasks, including deblurring, edges-to-handbags translation, and depth-to-RGB conversion. Notably, for handbag generation, our approach yields significantly more diverse outputs with varied colors and textures.

## 2 Background

**Notations** Let $\pi_T$, $\pi_0$, and $\pi_{0T}$ represent the base distribution, the target distribution, and the joint distribution of them respectively. $\pi_{\text{cond}}$ and $\pi_{\text{data}}$ represent the distributions of the input and output data. Let $p$ be the distribution of a diffusion process; we denote its marginal distribution at time $t$ by $p_t$, the conditional distribution at time $t$ given the state at time $s$ by $p_{t|s}$, and the distribution at time $t$ given the states at times $0$ and $T$ by $p_{t|0,T}$, i.e., the transition kernel of a bridge.

### 2.1 Denoising Diffusion Bridge Models

DDBMs [18] extend diffusion models to translate between two arbitrary distributions $\pi_0$ and $\pi_T$ given samples from them. Consider a reference process given by:

$$dX_t = f_t X_t dt + g_t dW_t, \tag{1}$$

whose transition kernel is given by $q_{t|0}(x_t|x_0) = \mathcal{N}(x_t; a_t x_0, \sigma_t^2 \mathbb{I})$. This process can be conditioned (or "pinned") at both an initial point $x_0$ and a terminal point $x_T$ to construct a diffusion bridge. Under mild assumptions, the pinned process is given by Doob's $h$-transform [29]:

$$\mathrm{d}X_t = \{f_t X_t + g_t^2 \nabla_{X_t} \log p_{T|t}(x_T|X_t)\}\mathrm{d}t + g_t \mathrm{d}W_t \tag{2}$$

where $\nabla_{X_t} \log p_{T|t}(x_T \mid X_t) = \frac{(a_t/a_T)x_T - X_t}{\sigma_t^2(\mathrm{SNR}_t/\mathrm{SNR}_T - 1)}$ and $\mathrm{SNR}_t := a_t^2/\sigma_t^2$ [18]. Eq. (2) is a stochastic process that transport from $p_0 = \pi_0$ and $p_t = \pi_t$, which is a valid bridge process. To sample from the conditional distribution $p(x_0|x_T)$, we can solve the reverse SDE or probability flow ODE from $t = T$ to $t = 0$ [18]:

$$\mathrm{d}X_t = \{f_t X_t + g_t^2(s - h)\}\mathrm{d}t + g_t \mathrm{d}W_t, \tag{3}$$

$$\mathrm{d}X_t = \{f_t X_t + g_t^2(s - \frac{1}{2}h)\}\mathrm{d}t, \tag{4}$$

where $X_T = x_T$, $s = \nabla_{X_t} \log p_{T|t}(x_T|X_t)$, $h = \nabla_{X_t} \log p_{t|T}(X_t|x_T)$. Generally, the score $\nabla_{x_t} \log p_{t|T}(x_t|x_T)$ in Eqs. (3) and (4) is intractable. However, it can be effectively estimated by denoising bridge score matching. Let $(x_0, x_T) \sim \pi_{0,T}(x_0, x_T)$, $x_t \sim p_{t|0,T}(x_t|x_0, x_T)$, $t \sim \mathcal{U}(0, T)$, and $\omega(t)$ be non-zero loss weighting term of any choice, then the score $\nabla_{x_t} \log p_{T|t}(x_T|x_t)$ can be approximated by a neural network $s_\theta(x_t, x_T, t)$ with denoising bridge score matching objective [18]:

$$\mathcal{L}(\theta) = \mathbb{E}\left[w(t)\|s_\theta(X_t, x_T, t) - \nabla_{x_t} \log p_{t|0,T}(X_t \mid x_0, x_T)\|^2\right]. \tag{5}$$

where $\mathbb{E}$ dentotes expectation over $x_t \sim p_{t|0,T}(x_t, x_0)$, $(x_0, x_T) \sim \pi_{0,T}$, $t \sim \mathcal{U}(0, T)$.

## 2.2 Diffusion Bridge Implicit Models

The transition kernel of the bridge process in Eq. (2) is given by [18, 42]:

$$p(x_t|x_0, x_T) = \mathcal{N}(x_t; \alpha_t x_0 + \beta_t x_T, \gamma_t^2 \mathbb{I}) \tag{6}$$

where $\alpha_t = a_t(1 - \frac{\mathrm{SNR}_T}{\mathrm{SNR}_t})$, $\beta_t = \frac{a_t}{a_T}\frac{\mathrm{SNR}_T}{\mathrm{SNR}_t}$, $\gamma_t^2 = \sigma_t^2(1 - \frac{\mathrm{SNR}_T}{\mathrm{SNR}_t})$. Suppose we sample in reverse time on the discretized timesteps $0 = t_0 < t_1 < \cdots t_{N-1} < t_N = T$. Then we can sample $x_0$ by the initial value $x_T$ and the updating rule:

$$x_{t_n} = \alpha_{t_n} x_T + \beta_{t_n} \hat{x}_0^\theta + \sqrt{\gamma_{t_n}^2 - \rho_{t_n}^2}\frac{x_{t_{n+1}} - \alpha_{t_{n+1}} x_T - \beta_{t_{n+1}} \hat{x}_0^\theta}{\gamma_{t_{n+1}}} + \rho_{t_n}\epsilon, \quad \epsilon \sim \mathcal{N}(0, \mathbb{I}). \tag{7}$$

where $\hat{x}_0^\theta(x_t, x_T, t)$ has the relation with the score function:

$$s_\theta(x_t, x_T, t) = -\frac{x_t - \alpha_t x_T - \beta_t \hat{x}_0^\theta(x_t, x_T, t)}{\gamma_t^2} \tag{8}$$

# 3 Have the bridge paths been fully explored?

Given the forward process defined in Eq. (1), diffusion bridge models [18, 42, 12, 8] utilize Doob's $h$-transform to construct a corresponding bridge process (Eq. (2)). While the resulting process effectively bridges the initial distribution $\pi_T$ and the target distribution $\pi_0$, such diffusion bridge approaches exhibit several limitations.

- **Parameter coupling.** Notice that the parameters $a_t$ and $\sigma_t$ are convolved in the transition kernel (Eq. (6)). Such coupling is unnecessary and decoupling those parameters is helpful for searching the 'best' bridge path.

- **Limited design space.** Despite Eq. (2) provides an infinite number of bridge paths by tuning $a_t$ and $\sigma_t$, but the space of bridge paths is still artificially restricted.

In contrast, the stochastic interpolants [1] framework allows a larger design space of bridge path with more decoupled parameters. Specifically, stochastic interpolants build a bridge path directly via the flow map:

$$\phi_t = \alpha_t x_0 + \beta_t x_T + \gamma_t z \tag{9}$$

where $z \sim \mathcal{N}(0, \mathbb{I})$. Eq. (9) builds a transport with $\pi_0$ and $\pi_T$ as boundary conditions if the kernel parameters satisfy [1]:

- $\alpha_0 = \beta_T = 1$ and $\alpha_T = \beta_0 = \gamma_0 = \gamma_1 = 0$;
- $\alpha_t, \beta_t, \gamma_t > 0$ for $t \in (0, T)$.

The transition kernel of the stochastic interpolants in Eq. (9) is a Gaussian distribution: $\mathcal{N}(x_t; \alpha_t x_0 + \beta_t x_T, \gamma_t^2 \mathbb{I})$. Unlike DDBM, which is parameterized by only two variables $a_t$ and $\sigma_t$, stochastic interpolants introduce decoupled parameters $\alpha_t$, $\beta_t$, and $\gamma_t$, offering a more flexible and expressive design space for constructing bridge paths.

A detailed discussion on the rationale behind the choices of $\alpha_t, \beta_t$, and and an ablation study on the shape of $\gamma_t$ is provided in App. E. Notably, the DDBM-VP and DDBM-VE models presented in [18] can be considered as special cases by choosing different $\alpha_t$, $\beta_t$, and $\gamma_t$, see App. D for more details. In the experiments, we limit the scope to linear transition kernels and set $T = 1$, i.e., $p_{t|0,T}(x_t|x_0, x_T) = \mathcal{N}(x_t; (1-t)x_0 + t x_1, 4\gamma_{\max}^2 t(1-t)\mathbb{I})$.

> Stochastic interpolants expands the space of bridge paths and leads to decoupled parameters compared to DDBM and DBIM.

## 4 Has the sampler space been fully explored?

For diffusion models, EDM [17] demonstrated that the design of training and sampling schemes could be decoupled to significantly improve results. We now explore whether a similar decoupling is possible for bridge models, and what freedom we have to improve sampling quality with a given trained model.

### 4.1 Endpoint-Conditioning for Stochastic Interpolants (ECSI)

Given transition kernel $p_{t|0,T}(x_t \mid x_0, x_T) = \mathcal{N}(x_t; \alpha_t x_0 + \beta_t x_T; \gamma_t \mathbb{I})$, we can identify the training objective 11, reverse sampling SDEs (Eq. (10)), as demonstrated in Proposition 4.1, see App. C for the proof.

**Proposition 4.1** (Endpoint-conditioned Stochastic Interpolants)**.** *Suppose the transition kernel of a diffusion bridge process is given by $p_{t|0,T}(x_t \mid x_0, x_T) = \mathcal{N}(x_t; \alpha_t x_0 + \beta_t x_T, \gamma_t^2 \mathbb{I})$, then the evolution of conditional probability $q_t(X_t|x_T)$ is given by the SDE:*

$$dX_t = b(t, X_t, x_T)dt + \sqrt{2\epsilon_t}dW_t, \tag{10}$$

*where $b(t, x_t, x_T) = \dot{\alpha}_t \hat{x}_0 + \dot{\beta}_t x_T + (\dot{\gamma}_t + \frac{\epsilon_t}{\gamma_t})\hat{z}_t$, $\hat{x}_0 = \mathbb{E}[x_0 \mid x_t, x_T]$, $\hat{z}_t =: (x_t - \alpha_t \hat{x}_0 - \beta_t x_T)/\gamma_t$. Besides, $\hat{x}_0$ can be approximated by neural networks $\hat{x}_0^\theta$ by minimizing a regression objective with the observed $x_0, x_T$ as targets,*

$$\mathcal{L}_0[\hat{x}_0^\theta] = \int_0^T \mathbb{E}[\|\hat{x}_0^\theta(t, x_t, x_T) - x_0\|_2^2]dt \tag{11}$$

*where $\mathbb{E}$ denotes an expectation over $(x_0, x_T) \sim \pi(x_0, x_T)$ and $x_t \sim p_t(x_t \mid x_0, x_T)$.*

**Relation to Stochastic Interpolants (SI)**. Both SI and ECSI in Prop. 4.1 can be seen as special cases of Conditioned SI. A key advantage of ECSI is its efficiency: while SI need to estimate two terms: $\mathbb{E}[x_0 \mid x_0]$ and $\mathbb{E}[x_1 \mid x_t]$, ECSI only estimate $\mathbb{E}[x_0 \mid x_t, x_1]$. A detailed comparison was demonstrated in App. B.

For training, we found that we could define an expanded space of bridge paths in terms of $\alpha_t, \beta_t, \gamma_t$, where $\gamma_t$ apparently controlled the *stochasticity* of the path. For sampling, we see from the proposition above that the sampling design space is expanded even further, as the sampling dynamics depend on $\alpha_t, \beta_t, \gamma_t$ and $\epsilon_t$, where $\epsilon_t$ appears as an additional degree of freedom to control stochasticity.

**Training.** Eq. (11) provides the training objective of the denoiser $\hat{x}_0^\theta(t, x_t, x_T)$. In the implementation, we include additional preconditioning as DDBM [18] and DBIM [42], see App. G for more details.

**Sampling.** We can generate samples from the conditional distribution $q_{0|T}(x_0 \mid x_T)$ by solving the stochastic differential equation in Eq. (10) from $t = T$ to $t = 0$.

## 4.2 Existing samplers are a strict subset of ECSI samplers

We now show that existing samplers implement a strict subset of the ECSI samplers, see Figure 1.

**DDBM sampler.** When $\epsilon_t = 0$, Eq. (10) reduces to a deterministic ODE. Setting $\epsilon_t = \gamma_t \dot{\gamma}_t - \frac{\dot{\alpha}_t}{\alpha_t}\gamma_t^2$ recovers the sampling SDE used in DDBM [18]. However, DDBM only provides a single reverse SDE and a single corresponding reverse ODE; it does not explore alternative choices of $\epsilon_t$.

**DBIM sampler.** For small enough $\Delta t$ and $\gamma_{t-\Delta t}^2 - 2\epsilon_t \Delta t > 0$, the sampling SDE can be discretized as:

$$x_{t-\Delta t} \approx \alpha_{t-\Delta t}\hat{x}_0 + \beta_{t-\Delta t}x_T + \tilde{z} \tag{12}$$

where $\bar{z}_t \sim \mathcal{N}(0, \mathbb{I})$, $\tilde{z} = \sqrt{\gamma_{t-\Delta t}^2 - 2\epsilon_t \Delta t}\hat{z}_t + \sqrt{2\epsilon_t \Delta t}\bar{z}_t$. Eq. (12) recover the DBIM sampler. Note that the condition $\gamma_{t-\Delta t}^2 - 2\epsilon_t \Delta t > 0$ limits the design space of samplers. For example, our best result in the experiments is achieved by setting $\alpha_t = 1 - t$, $\gamma_t = \frac{\gamma_{\max}^2}{4}t(1-t)$ and $\epsilon_t = \gamma_t\dot{\gamma}_t - \frac{\dot{\alpha}_t}{\alpha_t}\gamma_t^2$, DBIM sampler fails under this setting since $\gamma_{t-\Delta t}^2 - 2\epsilon_t \Delta t > 0$ cannot be guaranteed all the time.

**I²SB sampler.** When $2\epsilon_t \Delta t = \gamma_{t-\Delta t}^2 - \beta_{t-\Delta t}^2\gamma_t^2/\beta_t^2$, the coefficient of $x_T$ in Eq. (12) vanishes. This special case corresponds to the Markovian bridge introduced in [42], and notably allows us to recover the sampling procedure of I2SB [21]. We provide a detailed derivation of this connection in Appendix D. The design space of the I²SB sampler is also limited, as it can be interpreted as a special case of the DBIM sampler.

> Endpoint-Conditioned Stochastic Interpolants (Prop. 4.1) identify a class of sampling SDEs that share the same marginal distribution, but offer greater flexibility and a broader design space for sampler construction compared to DDBM, DBIM, and I2SB.

## 4.3 Our implementation

Our sampler based on Euler's discretization of the sampling SDE in Eq. (10):

$$x_{t-\Delta t} \approx x_t - b(t, x_t, x_T)\Delta t + \sqrt{2\epsilon_t \Delta t}\bar{z}_t, \tag{13}$$

We set $\epsilon_t = \eta(\gamma_t\dot{\gamma}_t - \frac{\dot{\alpha}_t}{\alpha_t}\gamma_t^2)$, where $\eta \in (0, 1)$ is an interpolation parameter. This formulation provides continuous control over the sampling process, ranging from purely deterministic ODE sampling ($\eta = 0$) to fully stochastic SDE sampling ($\eta = 1$). In our implementation, we let $\epsilon_t = 0$ for the last two steps, Eq. (12) gets reduced to: $x_{t-\Delta t} \approx \alpha_{t-\Delta t}\hat{x}_0 + \beta_{t-\Delta t}x_T + \gamma_{t-\Delta t}\hat{z}_t$. For other steps, we apply Eq. (13) and let $\epsilon_t = \eta(\gamma_t\dot{\gamma}_t - \frac{\dot{\alpha}_t}{\alpha_t}\gamma_t^2)$, where $\eta$ is a constant. Putting all ingredients together leads to our sampler outlined in Algorithm 1.

**Algorithm 1** ECSI Sampler

1: **Input:** $D_\theta(x_t, x_T, t)$, timesteps $\{t_j\}_{j=0}^N$, distribution $\pi_{\text{cond}}$, schedule $\alpha_t, \beta_t, \gamma_t, \epsilon_t, b$
2: Sample $x_T \sim \pi_{\text{cond}}, n_0 \sim \mathcal{N}(0, b^2\mathbb{I})$
3: $x_N = x_T + n_0$
4: **for** $i = N$ **to** $1$ **do**
5: $\quad \hat{x}_0 = D_\theta(x_i, x_T, t_i), \quad \hat{z}_i = (x_i - \alpha_{t_i}\hat{x}_0 - \beta_{t_i}x_N)/\gamma_{t_i}$
6: $\quad$ **if** $i \geq 2$ **then**
7: $\quad\quad$ Sample $\bar{z}_i \sim \mathcal{N}(0, \mathbb{I})$
8: $\quad\quad d_i = \dot{\alpha}_{t_i}\hat{x}_0 + \dot{\beta}_{t_i}x_N + (\dot{\gamma}_{t_i} + \epsilon_{t_i}/\gamma_{t_i})\hat{z}_i$
9: $\quad\quad x_{i-1} = x_i + d_i(t_i - t_{i-1}) + \sqrt{2\epsilon_{t_i}(t_i - t_{i-1})}\,\bar{z}_i$
10: $\quad$ **else**
11: $\quad\quad x_{i-1} = \alpha_{t_{i-1}}\hat{x}_0 + \beta_{t_{i-1}}x_N + \gamma_{t_{i-1}}\hat{z}_i$
12: $\quad$ **end if**
13: **end for**

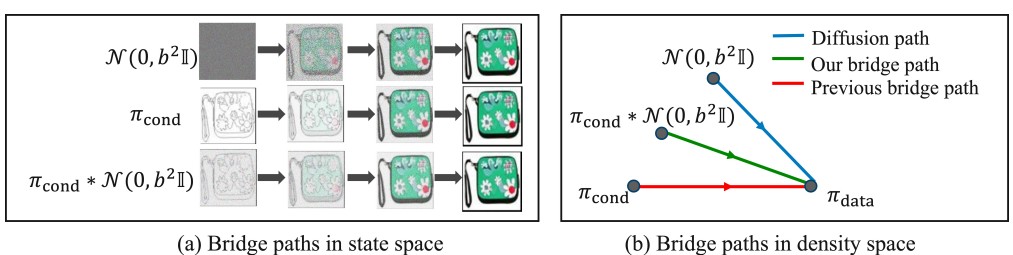

(a) Bridge paths in state space      (b) Bridge paths in density space

Figure 3: Modifying the base distribution corresponds to a lossy compression of the input that leads to a 'trade-off' between unconditional diffusion and diffusion bridge models.

## 5 Is there any benefit to modifying the starting point of a bridge?

We expanded the paths in distribution space connecting a base and target distribution, but so far left the endpoints fixed. While the target distribution should remain fixed, we could, in principle, modify the base distribution. At first glance this seems counter-intuitive - because of the data processing inequality we can only lose information about the target by modifying the base distribution. Hence, this angle has not been explored in the bridge literature. However, we found a surprising result - modifying the base distribution can help significantly. The situation is analogous to the benefits of lossy compression in VAEs [6]. Information in the base distribution is not necessarily helpful, so by modifying the base distribution (which destroys some information) the model can align better with natural factors of variation.

### 5.1 Low conditional diversity in one-to-many translations

In our experiments (see Sec. 6), we observe that existing diffusion bridge models tend to produce low-diversity outputs under fixed conditioning. For instance, when generating handbags from a single edge map, the model is expected to produce varied outputs in terms of color, texture, and fine details. However, we find that current bridge models generate visually similar images across different sampling runs, despite the injection of different noise realizations during the diffusion process.

To address the issue of low output diversity, we propose modifying the base distribution used in the bridge model. Prior works [18, 1] typically treat the base distribution $\pi_T$ as equivalent to the input data distribution, denoted $\pi_{\text{cond}}$. In contrast, our approach introduces a controlled perturbation by redefining the base distribution as $\pi_T = \pi_{\text{cond}} * \mathcal{N}(0, b^2\mathbb{I})$, where $b$ is a constant that governs the magnitude of noise added to the input distribution. This modification enables greater diversity in the generated outputs while maintaining conditional alignment.

Intuitively, this modification can be interpreted as a trade-off between standard diffusion models and traditional diffusion bridge models. As illustrated in Fig. 3, diffusion models typically generate samples starting from pure Gaussian noise, while diffusion bridge models begin sampling from fully conditioned inputs, such as edge maps. Our approach introduces an intermediate regime by

Table 2: Validation of our sampler via DDBM pretrained VP model (Evaluated by FID), where $\epsilon_t = 0.3(\gamma_t \dot{\gamma}_t - \frac{\dot{\alpha}_t}{\alpha_t} \gamma_t^2)$.

| Sampler | Edges→Handbags ($64 \times 64$) | | | DIODE-Outdoor ($256 \times 256$) | | |
|---|---|---|---|---|---|---|
| | NFE=5 | NFE=10 | NFE=20 | NFE=5 | NFE=10 | NFE=20 |
| DDBM [18] | 317.22 | 137.15 | 46.74 | 328.33 | 151.93 | 41.03 |
| DBIM [42] | 3.60 | 2.46 | 1.74 | 14.25 | 7.98 | 4.99 |
| ECSI (Ours) | **2.36** | **2.25** | **1.53** | **10.87** | **6.83** | **4.12** |

sampling from noisy conditioned inputs, thereby blending the benefits of both paradigms—preserving conditional guidance while enhancing output diversity.

> Modifying the base distribution with lossy compression can significantly improve the conditional diversity of the generated images.

## 5.2 How to measure the conditional diversity?

While existing metrics like FID implicitly capture the *unconditional* diversity of generated images, we need to capture the diversity of outputs (e.g. color images) for a single input image (a black and white edge map). To measure the conditional diversity, we will adopt Vendi Score (VS) [11] as a metric. Besides, We propose the Average Feature Distance (AFD) metric to quantify the conditional diversity among generated images. Initially, we select a group of source images $\{x_T^{(i)}\}_{i=1}^M$. For each $x_T^{(i)}$, we then generate $L$ distinct target samples. The $j$-th generated sample corresponding to the $i$-th source image is denoted by $y_{ij}$. Then the AFD is calculated as follows:

$$\text{AFD} = \frac{1}{M} \sum_{i=1}^M \frac{1}{L^2 - L} \sum_{k,l=1,k\neq l}^L \|F(y_{ik}) - F(y_{il})\| \tag{14}$$

where $F(\cdot)$ is a function that extracts the features of images, and $\|\cdot\|$ represents Euclidean norm. Intuitively, a larger AFD indicates the better conditional diversity. Here, $F(x)$ can be $x$ to evaluate the diversity directly in the pixel space. Alternatively, $F(\cdot)$ can be defined using the Inception-V3 model to assess the diversity in the latent space. In our experiments, we use AFD in latent space. Furthermore, we provide additional justification for the validity of our proposed metric in App. A.

**A comparison between AFD and VS**. Both AFD and the VS quantify diversity in the feature space of images, using features extracted from the Inception-V3 model. AFD measures the average pairwise Euclidean distance between feature vectors, making it sensitive to outliers. In contrast, the Vendi Score evaluates diversity by computing the effective number of unique feature patterns, based on the eigenvalues of the similarity matrix, emphasizing the overall structural diversity of the feature set. These metrics are complementary, capturing different aspects of diversity.

## 6 Experiments

In this section, we demonstrate how greatly expanding the space of bridge paths with ECSI leads to significantly improved performance for I2I translation tasks, in terms of sample efficiency, image quality and conditional diversity. We evaluate on I2I translation tasks on Edges→Handbags [16] scaled to $64 \times 64$ pixels and DIODE-Outdoor scaled to $256 \times 256$ [37], and Deblurring on ImageNet dataset [9]. For evaluation metrics, we use Fréchet Inception Distance (FID) [13] for all experiments, and additionally measure Inception Scores (IS) [3], Learned Perceptual Image Patch Similarity (LPIPS) [41], Mean Square Error (MSE), following previous works [42, 18]. In addition, we use VS and AFD, Eq. 14, to measure conditional diversity. Further details of the experiments and design guidelines are provided in Appendix G and E.

**Sampler**. We evaluate different sampling algorithms in Fig. 4 (a), the results demonstrate that setting $\epsilon_t = 0$ and using Eq. (12) for the last 2 steps can significantly improve sampled image

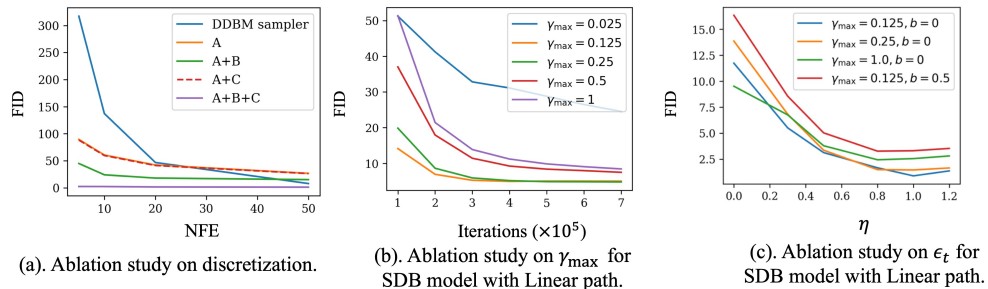

(a). Ablation study on discretization.

(b). Ablation study on $\gamma_{\max}$ for SDB model with Linear path.

(c). Ablation study on $\epsilon_t$ for SDB model with Linear path.

Figure 4: Ablation studies on discretization, $\gamma_{\max}$ and $\epsilon_t$. (a). We evaluate different discretization schemes on Edges2handbags ($64 \times 64$) dataset using DDBM-VP pretrained model, A represents simple Euler discretization in Eq. (13), B reprents setting $\epsilon_t = 0$ for the last 2 steps, C represents using Eq. (12) for $\epsilon_t = 0$. (b). Ablation study on $\gamma_{\max}$ evaluated by DIODE ($64 \times 64$) dataset. (c). Ablation study on $\epsilon_t$ through our ECSI model with Linear path on Edges2handbags ($64 \times 64$) dataset, where $\epsilon_t = \eta(\gamma_t \dot{\gamma}_t - \frac{\dot{\alpha}_t}{\alpha_t} \gamma_t^2)$.

Table 3: Quantitative results in the I2I translation task Edges2handbags ($64 \times 64$) and DIODE ($256 \times 256$) datasets. Our results were achieved by Linear transition kernel and setting $\eta = 1$.

| Model | NFE | Edges→handbags ($64 \times 64$) | | | | DIODE-Outdoor ($256 \times 256$) | | | |
| | | FID ↓ | IS ↑ | LPIPS ↓ | MSE | FID ↓ | IS ↑ | LPIPS ↓ | MSE |
|---|---|---|---|---|---|---|---|---|---|
| Pix2Pix [16] | 1 | 74.8 | 3.24 | 0.356 | 0.209 | 82.4 | 4.22 | 0.556 | 0.133 |
| DDIB [36] | $\geq 40^\dagger$ | 186.84 | 2.04 | 0.869 | 1.05 | 242.3 | 4.22 | 0.798 | 0.794 |
| SDEdit [25] | $\geq 40$ | 26.5 | 3.58 | 0.271 | 0.510 | 31.14 | 5.70 | 0.714 | 0.534 |
| Rectified Flow [22] | $\geq 40$ | 25.3 | 2.80 | 0.241 | 0.088 | 77.18 | 5.87 | 0.534 | 0.157 |
| I$^2$SB [21] | $\geq 40$ | 7.43 | 3.40 | 0.244 | 0.191 | 9.34 | 5.77 | 0.373 | 0.145 |
| DDBM [18] | 118 | 1.83 | 3.73 | 0.142 | 0.040 | 4.43 | 6.21 | 0.244 | 0.084 |
| DBIM [42] | 20 | 1.74 | 3.64 | 0.095 | 0.005 | 4.99 | 6.10 | 0.201 | 0.017 |
| ECSI ($\gamma_{\max} = 0.125$) | 5 | **0.89** | 4.10 | 0.049 | 0.024 | 12.97 | 5.49 | 0.269 | 0.074 |
| | 10 | **0.67** | 4.11 | 0.045 | 0.024 | 10.12 | 5.56 | 0.255 | 0.076 |
| | 20 | **0.56** | 4.11 | 0.044 | 0.024 | 8.62 | 5.62 | 0.248 | 0.078 |
| ECSI ($\gamma_{\max} = 0.25$) | 5 | 1.46 | **4.21** | **0.040** | 0.016 | **4.16** | 5.83 | **0.104** | 0.029 |
| | 10 | 1.38 | **4.22** | **0.038** | 0.017 | **3.44** | 5.86 | **0.098** | 0.029 |
| | 20 | 1.40 | **4.20** | **0.038** | 0.017 | **3.27** | 5.85 | **0.094** | 0.029 |

quality compared with simple Euler discretization and DDBM sampler. Furtheremore, By specifically designing noise control during sampling, our sampler surpasses the sampling results by DDBM and DBIM with the same pretrained model. The results are demonstrated in Table 2. We set the number of function evaluations (NFEs) from the set $[5, 10, 20]$.

**Bridge paths**. We introduced an extensive bridge design space and begin by focusing on linear transition paths with different strength of maximum stochasticity, i.e., $p_{t|0,T}(x_t|x_0, x_T) = \mathcal{N}(x_t; (1-t)x_0 + tx_T, \frac{1}{4}\gamma_{\max}^2 t(1-t)\mathbb{I})$. We conducted detailed ablation studies on $\gamma_{\max}$ and $\eta$ for the Linear path on DIODE ($64 \times 64$) dataset, as shown in Fig. 4 (b) and (c). The optimal values for $\gamma_{\max}$ were found to be $0.125$ and $0.25$, while the best performance for $\eta$ was achieved with $\eta = 0.8$ and $\eta = 1.0$. Performance deteriorates when either parameter is too small or too large. Based on the results of these ablation studies, we further trained ECSI models on the Edges2handbags ($64 \times 64$) and DIODE ($256 \times 256$) datasets by taking $\gamma_{\max} \in \{0.125, 0.5\}$ and setting $\eta = 1.0$. The results are presented in Table 3. Our models establish a new benchmark for image quality, as evaluated by FID, IS and LPIPS. Despite our models having slightly higher MSEs compared to the baseline DDBM and DBIM, we believe that a larger MSE indicates that the generated images are distinct from their references, suggesting a richer diversity.

**Modifying base distribution**. Through controlling noise in the base distribution, we achieved a more diverse set of sample images, while this diversity comes at the cost of slightly higher FID scores and slower sampling speed. We show generated images in Fig. 5. More visualization can be found in Appendix I, which shows that by introducing booting noise to the input data distribution, the model can generate samples with more diverse colors and textures. Further quantitative results are presented

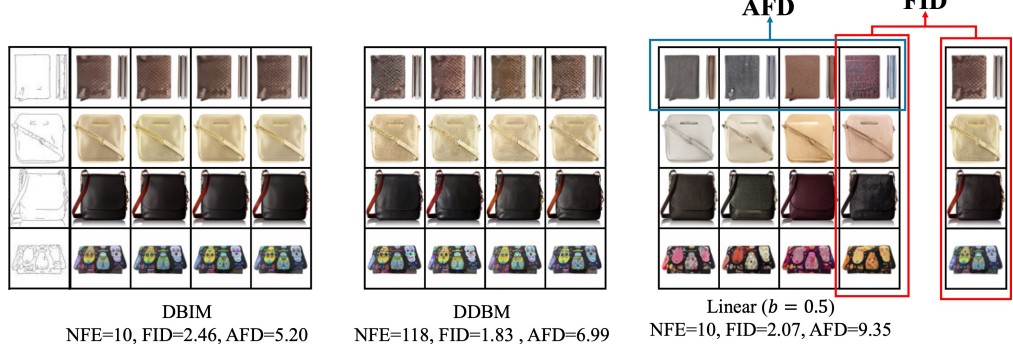

Figure 5: Visualization of conditional diversity via sampled images in a one-to-many translation task. While FID measures diversity within columns, AFD evaluates diversity across rows. The visualization further proved the effectiveness of AFD. More sampled images can be found in Appendix I.

Table 4: Quantitative results for Different denoisers and samplers on Edges2handbags ($64 \times 64$). Our baseline is achieved by DDBM pretrained checkpoint and DBIM sampler.

| Method | FID ↓ | | | AFD ↑ | | | VS ↑ | | |
|---|---|---|---|---|---|---|---|---|---|
| | NFE=5 | NFE=10 | NFE=20 | NFE=5 | NFE=10 | NFE=20 | NFE=5 | NFE=10 | NFE=20 |
| DDBM (pre) + DBIM sampler | 3.60 | 2.46 | 1.74 | 5.63 | 5.20 | 5.84 | 1.16 | 1.23 | 1.26 |
| A: DDBM (pre) + ECSI sampler | 2.36 | 2.25 | 1.53 | 5.11 | 5.70 | 6.04 | 1.15 | 1.20 | 1.23 |
| B: ECSI (pre) + ECSI sampler | 0.89 | 0.67 | 0.56 | 6.00 | 6.05 | 6.25 | 1.22 | 1.25 | 1.28 |
| B + Modified base density | 3.31 | 2.07 | 1.74 | 8.53 | 9.35 | 9.65 | 1.48 | 1.63 | 1.69 |

in Table 4, confirming that our model surpasses the vanilla DDBM in terms of image quality, sample efficiency, and conditional diversity.

**Deblurring on ImageNet Dataset**. We evaluate our models for Gaussian deblurring applying a Gaussian kernel with $\sigma = 10$ and Uniform deblurring, shown in Table 5. The results demonstrates that our ECSI models achieve much lower FID score.

# 7 Related Work

**Diffusion Bridge Models**. Diffusion bridges are faster diffusion processes that could learn the mapping between two random target distributions [39, 35], demonstrating significant potential in various areas, such as protein docking [32], mean-field game [20], I2I translation [21, 18]. According to different design philosophies, DBMs can be divided into two groups: bridge matching and stochastic interpolants. The idea of bridge matching was first proposed by Peluchetti et al. [28], and can be viewed as a generalization of score matching [34]. Based on this, diffusion Schrödinger bridge matching (DSBM) has been developed for solving Schrödinger bridge problems [35, 39]. In addition, Liu et al. [21] utilize bridge matching to perform image restoration tasks and noted benefits of noise empirically, the experiments shows the new model is more efficient and interpretable than score-based generative models [21]. Furthermore, our benchmark DDBM [18] achieve significant improvement for various I2I translation tasks, DBIM [42] improved the sampling algorithm for DDBM, significantly reducing sampling time while maintaining the same image quality.

**Image-to-Image Translations**. While diffusion models are strong at generating images, applying them to image-to-image (I2I) translation is more difficult due to artifacts in the output. DiffI2I improves quality and alignment with fewer diffusion steps [5]. In latent space, S2ST speeds up translation and reduces memory use [27]. Other methods improve guidance using features like frequency control [26, 15, 38]. A common challenge is that many models require joint training on both source and target domains, raising privacy concerns. Injecting-Diffusion tackles this by isolating shared content for unpaired translation [24]. SDDM improves interpretability by breaking down the score function across diffusion steps [30].

Table 5: Deblurring results with respect to different kernels, evaluated by FID on the 10k ImageNet ($256 \times 256$) validation subset. Our results are achieved by 20 NFEs.

| Kernel | DDRM | DDNM | Pallette | CDSB | I$^2$SB | ECSI (ours) |
|--------|------|------|----------|------|---------|-------------|
| Uniform | 9.9 | 3.0 | 4.1 | 15.5 | 3.9 | **1.11** |
| Gaussian | 6.1 | 2.9 | 3.1 | 7.7 | 3.0 | **0.41** |

## 8  Conclusion

We introduced Endpoint-Conditioned Stochastic Interpolants (ECSI)—an improved version of stochastic interpolants that adds endpoint conditioning, modifies the base distribution, and uses discretization to explore the design space of Diffusion Bridge Models (DBMs). We highlighted a key issue often overlooked: one-to-many image translation tasks lack conditional diversity. Our findings show that resolving this requires adjusting the starting distribution, not the path or sampler. ECSI sets new benchmarks in image quality, sampling efficiency, and conditional diversity on tasks like $64 \times 64$ edges2handbags, $256 \times 256$ DIODE-outdoor, and ImageNet deblurring.

**Limitations**. (i) We note that optimal path design may vary by task, leaving room for future refinement. (ii) Incorporating guidance techniques may further enhance model performance.

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

Table 6: Evaluation for generative models: ImageNet-1-mode, ImageNet-2-modes, ImageNet-5-modes, and ImageNet-10-modes.

| Model | ImageNet-1-mode | ImageNet-2-modes | ImageNet-5-modes | ImageNet-10-modes |
|-------|-----------------|------------------|------------------|-------------------|
| FID   | 58.30           | 57.34            | 57.78            | 57.26             |
| AFD   | 0               | 8.14             | 12.84            | 14.47             |

# A  AFD validation

In this section, we thoroughly validate the effectiveness of our proposed metric, AFD, for measuring conditional diversity and demonstrate its role as a complementary metric to FID. In unconditional generation scenarios, the FID is widely used to evaluate the diversity of generated images. While low FID scores generally indicate high diversity across the entire dataset, they do not necessarily imply high conditional diversity. For instance, we observed that samples generated by the DDBM model often lack diversity when conditioned on edge images, despite achieving very low FID scores. To address this limitation, we introduce the concept of conditional diversity and propose a corresponding metric to quantify it.

The first question is why FID failed to measure the conditional diversity. To illustrate the limitations of FID in capturing conditional diversity, consider an extreme case: if the images generated by a generative model are identical to a set of baseline images, the FID score can be very low since the two distributions are indistinguishable. However, this scenario does not reflect diversity within the conditional outputs.

To further support our point, we designed two classes of pseudo-generative models capable of controlling the diversity of the generated images, which are further validated by FID and AFD. The experiments are evaluated on Imagenet dataset [9].

## A.1  Pseudo-generative models by random selection

We designed four pseudo-generative models: ImageNet-1-mode, ImageNet-2-modes, ImageNet-5-modes, and ImageNet-10-modes. The experimental setup is as follows:

- We selected 11,000 samples from the ImageNet validation dataset, randomly choosing 11 images per class.

- From these, we designated 1,000 images as the "real" set, while the remaining images served as the source pool for the generative models.

- Each ImageNet-k-modes model simulates a generative process by randomly sampling images from a pool of $k$ distinct images within a given class.

We present sampled images in Fig. 6, where it is evident that the ImageNet-10-modes model generates images with the highest conditional diversity. To quantify this, we conducted experiments to calculate both FID and AFD for the four generative models. The results are summarized in Table 6. While the FID scores are nearly identical across all models, the AFD values increase as the conditional diversity of the generative models improves. This highlights that AFD is a more effective metric for capturing conditional diversity than FID.

## A.2  Pseudo-generative models by strong augmentation

Strong augmentation has been widely used in computer vision to generate synthetic data while preserving its underlying semantics [7, 40, 31, 4]. The intensity of augmentation can be adjusted, with higher intensities producing more diverse images. To further validate our proposed metric, AFD, as a measure of diversity, we construct pseudo-generative models using strong augmentation.

We selected 1,000 images from the ImageNet-1k dataset, one from each category. These images were subjected to data augmentation, specifically using ColorJitter, with varying magnitudes to enhance diversity. For each image, the augmentation was applied 16 times, creating an augmented dataset for

ImageNet-1-mode: FID=58.30, AFD=0

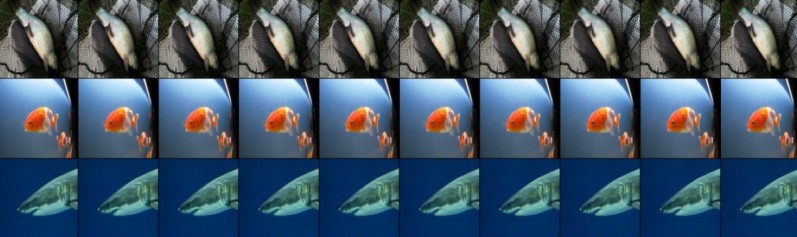

ImageNet-2-modes: FID=57.34, AFD=8.14

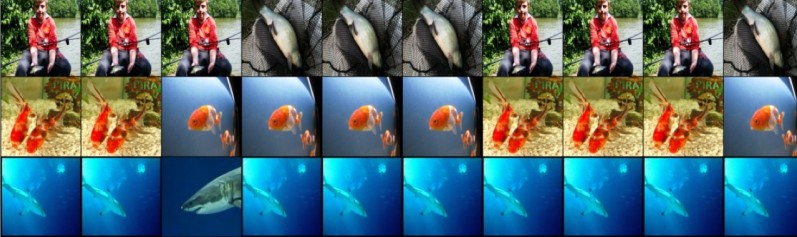

ImageNet-5-modes: FID=57.78, AFD=12.84

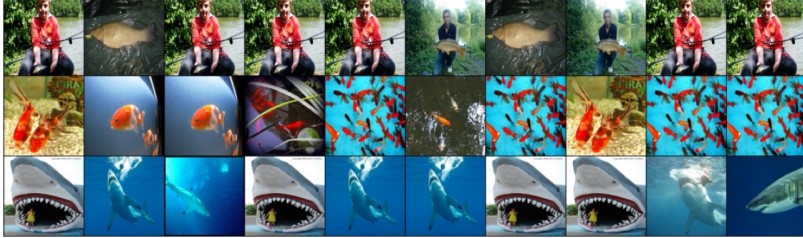

ImageNet-10-modes: FID=57.26, AFD=14.47

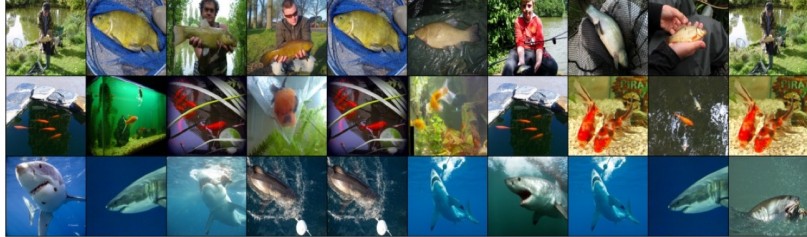

Figure 6: Sampled images from 4 generative models: ImageNet-1-mode, ImageNet-2-modes, ImageNet-5-modes, ImageNet-10-modes.

each magnitude setting. We then calculated the AFD for these augmented datasets to evaluate the relationship between dataset diversity (as influenced by augmentation magnitude) and the AFD value.

Table 7 summarizes the AFD results across various augmentation magnitude settings. The results show that as diversity increases, AFD values also rise, further confirming that the proposed AFD metric is a reliable indicator of image diversity.

## B  Relation to Stochastic Interpolants

Conditioned Stochastic Interpolants build a marginal probability path $p_{t|y}$ using a mixture of interpolating densities: $p_{t|y}(x) = \int p_t(x_t \mid x_0, x_1)\pi(x_0, x_1 \mid y)dx_0 dx_1$, where $\pi(x_0, x_1 \mid y)$ is a joint distribution with marginals $\pi_{0|y}(x_0 \mid y)$ and $\pi_{1|y}(x_1 \mid y)$. For linear interpolants given by: $X_t = \alpha_t X_0 + \beta_t X_1 + \gamma_t z$. The conditional kernel $p_t(x_t \mid x_0, x_1)$ is given by a Gaussian distribution:

Table 7: AFD results across different augmentation magnitudes

| Augmentation magnitude | 0.1 | 0.2 | 0.3 | 0.4 | 0.5 | 0.6 | 0.7 | 0.8 |
|---|---|---|---|---|---|---|---|---|
| AFD | 2.16 | 3.77 | 5.13 | 6.16 | 6.98 | 7.63 | 8.22 | 9.01 |
| FID | 0.20 | 2.95 | 7.02 | 11.62 | 16.33 | 20.84 | 25.12 | 28.89 |

$p_t(x_t \mid x_0, x_1) = \mathcal{N}(\alpha_t x_0 + \beta_t x_1, \sigma_t^2 I), \forall\, t \in [0, 1]$. Then we can sample from the conditional distribution $p_{0|y}(x_0 \mid y)$ by running a stochastic process $p_{t|y}(x_t \mid y)$ from time $t = 1$ to $t = 0$, which is given by the following SDE:

$$dx = b(t, x, y)dt + \sqrt{2\epsilon_t}dW_t, \quad x_1 \sim p_{1|y}, \tag{15}$$

where the drift term $b(t, x, y)$ is:

$$b(t, x, y) = \dot{\alpha}_t \mathbb{E}[x_0 \mid x, y] + \dot{\beta}_t \mathbb{E}[x_1 \mid x, y] + (\dot{\gamma}_t + \frac{\epsilon_t}{\gamma_t})\mathbb{E}[z \mid x, y] \tag{16}$$

As $y$ represents null conditioning, Eq. (15) recover the original sampler of Stochastic Interpolants. In the drift term, $\mathbb{E}[x_0 \mid x, y]$, $\mathbb{E}[x_1 \mid x, y]$ and $\mathbb{E}[z \mid x, y]$ are unknown, but we only need to estimate two of them, since

$$\mathbb{E}[x_t \mid x_t, y] = \alpha_t \mathbb{E}[x_0 \mid x_t, y] + \beta_t \mathbb{E}[x_1 \mid x_t, y] + \gamma_t \mathbb{E}[z \mid x_t, y] = x_t$$

We can further reduce the number of unknown term by endpoint-conditioning. Here as we replace condition $y$ to be endpoint $x_1$, the term $\mathbb{E}[x_1 \mid x, x_1] = x_1$. So we have:

$$b(t, x, y) = \dot{\alpha}_t \mathbb{E}[x_0 \mid x, x_1] + \dot{\beta}_t x_1 + (\dot{\gamma}_t + \frac{\epsilon_t}{\gamma_t})\mathbb{E}[z \mid x, x_1] \tag{17}$$

This is exactly the sampler for ECSI in Prop. 4.1. Therefore, both SI and ECSI can be seen as special cases of Conditioned SI. A key advantage of ECSI is its efficiency: while SI need to estimate two terms: $\mathbb{E}[x_0 \mid x_t]$ and $\mathbb{E}[x_1 \mid x_t]$, ECSI only estimates $\mathbb{E}[x_0 \mid x_t, y]$.

## C  Proofs

There are infinitely many pinned processes characterized by the Gaussian transition kernel $p_{t|0,T}(\mathbf{x}_t \mid \mathbf{x}_0, \mathbf{x}_T) = \mathcal{N}(\mathbf{x}_t; \alpha_t \mathbf{x}_0 + \beta_t \mathbf{x}_T, \gamma_t^2 \mathbb{I})$. Specifically, we formalize the pinned process as a linear Itô SDE, as presented in Lemma C.1.

**Lemma C.1.** *There exist a linear Itô SDE*

$$d\mathbf{X}_t = [f_t \mathbf{X}_t + s_t \mathbf{x}_T]dt + g_t d\mathbf{W}_t, \quad \mathbf{X}_0 = \mathbf{x}_0, \tag{18}$$

*where* $f_t = \frac{\dot{\alpha}_t}{\alpha_t}$, $s_t = \dot{\beta}_t - \frac{\dot{\alpha}_t}{\alpha_t}\beta_t$, $g_t = \sqrt{2(\gamma_t \dot{\gamma}_t - \frac{\dot{\alpha}_t}{\alpha_t}\gamma_t^2)}$, *that has a Gaussian marginal distribution* $\mathcal{N}\left(\mathbf{x}_t; \alpha_t \mathbf{x}_0 + \beta_t \mathbf{x}_T, \gamma_t^2 \mathbb{I}\right)$.

*Proof.* Let $\mathbf{m}_t$ denote the mean function of the given Itô SDE, then we have $\frac{d\mathbf{m}_t}{dt} = f_t \mathbf{m}_t + s_t \mathbf{x}_T$. Given the transition kernel, the mean function $\mathbf{m}_t = \alpha_t \mathbf{x}_0 + \beta_t \mathbf{x}_T$, therefore,

$$\dot{\alpha}_t \mathbf{x}_0 + \dot{\beta}_t \mathbf{x}_T = f_t(\alpha_t \mathbf{x}_0 + \beta_t \mathbf{x}_T) + s_t \mathbf{x}_T. \tag{19}$$

Matching the above equation:

$$f_t = \frac{\dot{\alpha}_t}{\alpha_t}, s_t = \dot{\beta}_t - \beta_t \frac{\dot{\alpha}_t}{\alpha_t}. \tag{20}$$

Further, For the variance $\gamma_t^2$ of the process, the dynamics are given by:

$$\frac{d\gamma_t^2}{dt} = 2f_t\gamma_t^2 + g_t^2. \tag{21}$$

Solving for $g_t^2$, we substitute $f_t = \frac{\dot{\alpha}_t}{\alpha_t}$:

$$g_t^2 = \frac{d\gamma_t^2}{dt} - 2\frac{\dot{\alpha}_t}{\alpha_t}\gamma_t^2 \tag{22}$$

Therefore,

$$g_t = \sqrt{2(\gamma_t\dot{\gamma}_t - \frac{\dot{\alpha}_t}{\alpha_t}\gamma_t^2)}. \tag{23}$$

$\square$

Given the pinned process (18), we can sample from the conditional distribution $p_{0|T}(\mathbf{x}_0|\mathbf{x}_T)$ by solving the reverse SDE or ODE from $t = T$ to $t = 0$:

$$d\mathbf{X}_t = \left[f_t\mathbf{X}_t + s_t\mathbf{x}_T - g_t^2\nabla_{\mathbf{X}_t}\log p_t(\mathbf{X}_t|\mathbf{x}_T)\right]dt + g_t d\mathbf{W}_t, \quad \mathbf{X}_T = \mathbf{x}_T, \tag{24}$$

$$d\mathbf{X}_t = \left[f_t\mathbf{X}_t + s_t\mathbf{x}_T - \frac{1}{2}g_t^2\nabla_{\mathbf{X}_t}\log p_t(\mathbf{X}_t|\mathbf{x}_T)\right]dt \quad \mathbf{X}_T = \mathbf{x}_T, \tag{25}$$

where the score $\nabla_{\mathbf{X}_t}\log p_t(\mathbf{X}_t|\mathbf{x}_T)$ can be estimated by score matching objective (5).

For dynamics described by ODE $d\mathbf{X}_t = \mathbf{u}_t dt$, we can identify the entire class of SDEs that maintain the same marginal distributions, as detailed in Lemma C.2. This enables us to control the noise during sampling by appropriately designing $\epsilon_t$.

**Lemma C.2.** *Consider a continuous dynamics given by ODE of the form: $d\mathbf{X}_t = \mathbf{u}_t dt$, with the density evolution $p_t(\mathbf{X}_t)$. Then there exists forward SDEs and backward SDEs that match the marginal distribution $p_t$. The forward SDEs are given by: $d\mathbf{X}_t = (\mathbf{u}_t + \epsilon_t\nabla\log p_t)dt + \sqrt{2\epsilon_t}d\mathbf{W}_t, \epsilon_t > 0$. The backward SDEs are given by: $d\mathbf{X}_t = (\mathbf{u}_t - \epsilon_t\nabla\log p_t)dt + \sqrt{2\epsilon_t}d\mathbf{W}_t, \epsilon_t > 0$.*

*Proof.* For the forward SDEs, the Fokker-Planck equations are given by:

$$\frac{\partial p_t(\mathbf{X}_t)}{\partial t} = -\nabla \cdot \left[(\mathbf{u}_t + \epsilon_t\nabla\log p_t)\, p_t(\mathbf{X}_t)\right] + \epsilon_t\nabla^2 p_t(\mathbf{X}_t) \tag{26}$$

$$= -\nabla \cdot \left[\mathbf{u}_t p_t(\mathbf{X}_t)\right] - \nabla \cdot \left[\epsilon_t(\nabla\log p_t)p_t(\mathbf{X}_t)\right] + \epsilon_t\nabla^2 p_t(\mathbf{X}_t) \tag{27}$$

$$= -\nabla \cdot \left[\mathbf{u}_t p_t(\mathbf{X}_t)\right] - \epsilon_t\nabla \cdot \left[\nabla p_t(\mathbf{X}_t)\right] + \epsilon_t\nabla^2 p_t(\mathbf{X}_t) \tag{28}$$

$$= -\nabla \cdot \left[\mathbf{u}_t p_t(\mathbf{X}_t)\right]. \tag{29}$$

This is exactly the Fokker-Planck equation for the original deterministic ODE $d\mathbf{X}_t = \mathbf{u}_t\, dt$. Therefore, the forward SDE maintains the same marginal distribution $p_t(\mathbf{X}_t)$ as the original ODE.

Now consider the backward SDEs, the Fokker-Planck equations become:

$$\frac{\partial p_t(\mathbf{X}_t)}{\partial t} = -\nabla \cdot \left[(\mathbf{u}_t - \epsilon_t\nabla\log p_t)\, p_t(\mathbf{X}_t)\right] - \epsilon_t\nabla^2 p_t(\mathbf{X}_t) \tag{30}$$

$$= -\nabla \cdot \left[\mathbf{u}_t p_t(\mathbf{X}_t)\right] + \nabla \cdot \left[\epsilon_t(\nabla\log p_t)p_t(\mathbf{X}_t)\right] - \epsilon_t\nabla^2 p_t(\mathbf{X}_t) \tag{31}$$

$$= -\nabla \cdot \left[\mathbf{u}_t p_t(\mathbf{X}_t)\right]. \tag{32}$$

This is again the Fokker-Planck equation corresponding to the original deterministic ODE $d\mathbf{X}_t = \mathbf{u}_t\, dt$. Therefore, the backward SDE also maintains the same marginal distribution $p_t(\mathbf{X}_t)$.

$\square$

**Lemma C.3.** *Let $(\mathbf{x}_0, \mathbf{x}_T) \sim \pi_0(\mathbf{x}_0, \mathbf{x}_T)$, $\mathbf{x}_t \sim p_t(\mathbf{x}|\mathbf{x}_0, \mathbf{x}_T)$, Given the transition kernel: $p(\mathbf{x}_t \mid \mathbf{x}_0, \mathbf{x}_T) = \mathcal{N}\left(\mathbf{x}_t; \alpha_t\mathbf{x}_0 + \beta_t\mathbf{x}_T, \gamma_t^2\mathbb{I}\right)$, if $\hat{\mathbf{x}}_0(\mathbf{x}_t, \mathbf{x}_T, t)$ is a denoiser function that minimizes the expected $L_2$ denoising error for samples drawn from $\pi_0(\mathbf{x}_0, \mathbf{x}_T)$:*

$$\hat{\mathbf{x}}_0(\mathbf{x}_t, \mathbf{x}_T, t) = \arg \min_{D(\mathbf{x}_t, \mathbf{x}_T, t)} \mathbb{E}_{\mathbf{x}_0, \mathbf{x}_T, \mathbf{x}_t} \left[\lambda(t)\|D(\mathbf{x}_t, \mathbf{x}_T, t) - \mathbf{x}_0\|_2^2\right], \tag{33}$$

*then the score has the following relationship with $\hat{\mathbf{x}}_0(\mathbf{x}_t, \mathbf{x}_T, t)$:*

$$\nabla_{\mathbf{x}_t} \log p_t(\mathbf{x}_t|\mathbf{x}_T) = \frac{\alpha_t\hat{\mathbf{x}}_0(\mathbf{x}_t, \mathbf{x}_T, t) + \beta_t\mathbf{x}_T - \mathbf{x}_t}{\gamma_t^2}. \tag{34}$$

*Proof.*

$$\mathcal{L}(D) = \mathbb{E}_{(\mathbf{x}_0, \mathbf{x}_T)\sim\pi_0(\mathbf{x}_0, \mathbf{x}_T)}\mathbb{E}_{\mathbf{x}_t\sim p_t(\mathbf{x}_t|\mathbf{x}_0, \mathbf{x}_T)}\|D(\mathbf{x}_t) - \mathbf{x}_0\|_2^2 \tag{35}$$

$$= \int_{\mathbb{R}^d}\int_{\mathbb{R}^d}\underbrace{\int_{\mathbb{R}^d} p_t(\mathbf{x}_t|\mathbf{x}_0, \mathbf{x}_T)\pi_0(\mathbf{x}_0, \mathbf{x}_T)\|D(\mathbf{x}_t) - \mathbf{x}_0\|_2^2\,\mathrm{d}\mathbf{x}_0}_{=:\mathcal{L}(D;\mathbf{x}_t, \mathbf{x}_T)}\,\mathrm{d}\mathbf{x}_T\mathrm{d}\mathbf{x}_t, \tag{36}$$

$$\mathcal{L}(D; \mathbf{x}_t, \mathbf{x}_T) = \int_{\mathbb{R}^d} p_t(\mathbf{x}_t|\mathbf{x}_0, \mathbf{x}_T)\pi_0(\mathbf{x}_0, \mathbf{x}_T)\|D(\mathbf{x}_t) - \mathbf{x}_0\|_2^2\,\mathrm{d}\mathbf{x}_0, \tag{37}$$

we can minimize $\mathcal{L}(D)$ by minimizing $\mathcal{L}(D; \mathbf{x}_t, \mathbf{x}_T)$ independently for each $\{\mathbf{x}_t, \mathbf{x}_T\}$ pair.

$$D^*(\mathbf{x}_t, \mathbf{x}_T) = \arg \min_{D(\mathbf{x}_t)} \mathcal{L}(D; \mathbf{x}_t, \mathbf{x}_T) \tag{38}$$

$$\mathbf{0} = \nabla_{D(\mathbf{x}_t, \mathbf{x}_T)}[\mathcal{L}(D; \mathbf{x}_t, \mathbf{x}_T)] \tag{39}$$

$$= \int_{\mathbb{R}^d} p_t(\mathbf{x}_t|\mathbf{x}_0, \mathbf{x}_T)\pi_0(\mathbf{x}_0, \mathbf{x}_T)2[D(\mathbf{x}, \mathbf{x}_T) - \mathbf{x}_0]\,\mathrm{d}\mathbf{x}_0 \tag{40}$$

$$= 2[D(\mathbf{x}_t, \mathbf{x}_T)\int_{\mathbb{R}^d} p_t(\mathbf{x}_t|\mathbf{x}_0, \mathbf{x}_T)\pi_0(\mathbf{x}_0, \mathbf{x}_T)\,\mathrm{d}\mathbf{x}_0 - \int_{\mathbb{R}^d} p_t(\mathbf{x}_t|\mathbf{x}_0, \mathbf{x}_T)\pi_0(\mathbf{x}_0, \mathbf{x}_T)\mathbf{x}_0\,\mathrm{d}\mathbf{x}_0] \tag{41}$$

$$= 2[D(\mathbf{x})p_t(\mathbf{x}_t, \mathbf{x}_T) - \int_{\mathbb{R}^d} p_t(\mathbf{x}_t|\mathbf{x}_0, \mathbf{x}_T)\pi_0(\mathbf{x}_0, \mathbf{x}_T)\mathbf{x}_0\,\mathrm{d}\mathbf{x}_0], \tag{42}$$

$$D^*(\mathbf{x}_t, \mathbf{x}_T) = \int_{\mathbb{R}^d} \frac{p_t(\mathbf{x}_t|\mathbf{x}_0, \mathbf{x}_T)\pi_0(\mathbf{x}_0, \mathbf{x}_T)\mathbf{x}_0}{p_t(\mathbf{x}_t, \mathbf{x}_T)}\,\mathrm{d}\mathbf{x}_0, \tag{43}$$

$$\nabla_{\mathbf{x}_t} \log p_t(\mathbf{x}_t|\mathbf{x}_T) = \frac{\nabla_{\mathbf{x}_t} p_t(\mathbf{x}_t, \mathbf{x}_T)}{p_t(\mathbf{x}_t, \mathbf{x}_T)} \tag{44}$$

$$= \frac{\int \nabla_{\mathbf{x}_t} p_t(\mathbf{x}_t|\mathbf{x}_T, \mathbf{x}_0)\pi_0(\mathbf{x}_0, \mathbf{x}_T)d\mathbf{x}_0}{p_t(\mathbf{x}_t, \mathbf{x}_T)} \tag{45}$$

$$= -\int \frac{\mathbf{x}_t - \alpha_t\mathbf{x}_0 - \beta_t\mathbf{x}_T}{\gamma^2}\frac{p_t(\mathbf{x}_t|\mathbf{x}_0, \mathbf{x}_T)\pi_0(\mathbf{x}_0, \mathbf{x}_T)}{p_t(\mathbf{x}_t, \mathbf{x}_T)}d\mathbf{x}_0 \tag{46}$$

$$= \frac{\alpha_t D^*(\mathbf{x}_t, \mathbf{x}_T) + \beta_t\mathbf{x}_T - \mathbf{x}_t}{\gamma^2}. \tag{47}$$

Thus we conclude the proof.

$\square$

Proof of Prop. 4.1.

Table 8: Specify design choices for different model families. In the implementation, $\sigma_t = t$ for EDM, $\sigma_t = t, a_t = 1$ for DDBM-VE, $\sigma_t = \sqrt{e^{\frac{1}{2}\beta_d t^2 + \beta_{\min} t} - 1}$ and $a_t = 1/\sqrt{e^{\frac{1}{2}\beta_d t^2 + \beta_{\min} t}}$ for DDBM-VP, where $\beta_d$ and $\beta_{\min}$ are parameters. We include details and proofs in Appendix D.

| | | I2SB | DDBM | DBIM | EDM | Ours |
|---|---|---|---|---|---|---|
| | $\alpha_t$ | $1 - \sigma_t^2/\sigma_T^2$ | $a_t(1 - a_T^2\sigma_t^2/(\sigma_t^2 a_t^2))$ | $a_t(1 - a_T^2\sigma_t^2/(\sigma_t^2 a_t^2))$ | 1 | $1 - t$ |
| Transition kernel | $\beta_t$ | $\sigma_t^2/\sigma_T^2$ | $a_T\sigma_t^2/(\sigma_t^2 a_t)$ | $a_T\sigma_t^2/(\sigma_t^2 a_t)$ | 0 | $t$ |
| | $\gamma_t^2$ | $\sigma_t^2(1 - \sigma_t^2/\sigma_T^2)$ | $\sigma_t^2(1 - a_T^2\sigma_t^2/(\sigma_t^2 a_t^2))$ | $\sigma_t^2(1 - a_T^2\sigma_t^2/(\sigma_t^2 a_t^2))$ | $\sigma_t^2$ | $\frac{\gamma_{\max}^2}{4}t(1-t)$ |
| Sampling SDEs | $\epsilon_t$ | $\frac{\gamma_{t-\Delta t}^2\beta_t^2 - \beta_{t-\Delta t}^2\gamma_t^2}{2\beta_t^2\Delta t}$ | $\eta(\gamma_t\dot{\gamma}_t - \frac{\dot{\alpha}_t}{\alpha_t}\gamma_t^2)$ $\eta = 0$ or $\eta = 1$ | $\begin{cases} \frac{\gamma_{t-\Delta t}^2}{2\Delta t}, & t = 0 \\ 0, & t \neq 0 \end{cases}$ | $\bar{\beta}_t\sigma_t^2$ - | $\eta(\gamma_t\dot{\gamma}_t - \frac{\dot{\alpha}_t}{\alpha_t}\gamma_t^2)$ $\eta \in [0,1]$ |
| Base distribution $\pi_T$ | | $\pi_{\text{cond}}$ | $\pi_{\text{cond}}$ | $\pi_{\text{cond}}$ | $\pi_{\text{cond}}$ | $\pi_{\text{cond}} * \mathcal{N}(0, b^2\mathbb{I})$ |
| Discretization | - | | Euler Eq. (12) | Euler Eq. (13) | Euler Eq. (12) | Heun - | Euler Eqs. (13) and (12) |

*Proof.* Recall Eqs. (24) (25) and Lemma C.2,

$$d\mathbf{X}_t = \left[\frac{\dot{\alpha}_t}{\alpha_t}\mathbf{x}_t + (\dot{\beta}_t - \frac{\dot{\alpha}_t}{\alpha_t}\beta_t)\mathbf{x}_T - (\gamma_t\dot{\gamma}_t - \frac{\dot{\alpha}_t}{\alpha_t}\gamma_t^2 + \epsilon_t)\nabla_{\mathbf{x}_t}\log p_t(\mathbf{x}_t|\mathbf{x}_T)\right]dt + \sqrt{2\epsilon_t}d\mathbf{w}_t. \quad (48)$$

$\square$

Next we take the reparameterized score in Eq. (34) into Eq. (48):

$$d\mathbf{X}_t = \left[\frac{\dot{\alpha}_t}{\alpha_t}\mathbf{X}_t + (\dot{\beta}_t - \frac{\dot{\alpha}_t}{\alpha_t}\beta_t)\mathbf{x}_T - (\gamma_t\dot{\gamma}_t - \frac{\dot{\alpha}_t}{\alpha_t}\gamma_t^2 + \epsilon_t)\frac{\alpha_t\hat{\mathbf{x}}_0 + \beta_t\mathbf{x}_T - \mathbf{X}_t}{\gamma_t^2}\right]dt + \sqrt{2\epsilon_t}d\mathbf{w}_t \quad (49)$$

$$= \left[\dot{\alpha}_t\hat{\mathbf{x}}_0 + \dot{\beta}_t\mathbf{x}_T - (\gamma_t\dot{\gamma}_t + \epsilon_t)\frac{\alpha_t\hat{\mathbf{x}}_0 + \beta_t\mathbf{x}_T - \mathbf{X}_t}{\gamma_t^2}\right]dt + \sqrt{2\epsilon_t}d\mathbf{w}_t \quad (50)$$

$$= \left[\dot{\alpha}_t\hat{\mathbf{x}}_0 + \dot{\beta}_t\mathbf{x}_T - (\dot{\gamma}_t + \frac{\epsilon_t}{\gamma_t})\frac{\alpha_t\hat{\mathbf{x}}_0 + \beta_t\mathbf{x}_T - \mathbf{X}_t}{\gamma_t}\right]dt + \sqrt{2\epsilon_t}d\mathbf{w}_t \quad (51)$$

$$= \left[\dot{\alpha}_t\hat{\mathbf{x}}_0 + \dot{\beta}_t\mathbf{x}_T - (\dot{\gamma}_t + \frac{\epsilon_t}{\gamma_t})\hat{\mathbf{z}}\right]dt + \sqrt{2\epsilon_t}d\mathbf{w}_t. \quad (52)$$

## D   Reframing previous methods in our framework

We draw a link between our framework and the diffusion bridge models used in DDBM.

### D.1   DDBM-VE

DDBM-VE can be reformulated in our framework as we set :

$$\alpha_t = s_t(1 - \frac{\sigma_t^2}{\sigma_T^2}), \beta_t = \frac{s_t\sigma_t^2}{s_1\sigma_T^2}, \gamma_t = \sigma_t s_t\sqrt{(1 - \frac{\sigma_t^2}{\sigma_T^2})} \quad (53)$$

*Proof.* In the origin DDBM paper, the evolution of conditional probability $q(\mathbf{x}_t|\mathbf{x}_T)$ has a time reversed SDE of the form:

$$d\mathbf{X}_t = \left[\bar{\mathbf{f}}_t(\mathbf{X}_t) - g_t^2\bar{\mathbf{h}}_t(\mathbf{X}_t) - g_t^2\mathbf{s}_t(\mathbf{X}_t)\right]dt + g_t d\hat{\mathbf{W}}_t, \quad (54)$$

and an associated probability flow ODE

$$d\mathbf{X}_t = \left[\bar{\mathbf{f}}_t(\mathbf{X}_t) - g_t^2\bar{\mathbf{h}}_t(\mathbf{X}_t) - \frac{1}{2}g_t^2\mathbf{s}_t(\mathbf{X}_t)\right]dt. \tag{55}$$

Compare Eqs. (54) and 55 with Lemma C.1. We only need to prove:

$$\bar{\mathbf{f}}_t(\mathbf{X}_t) - g_t^2\bar{\mathbf{h}}_t(\mathbf{X}_t) = f_t\mathbf{X}_t + s_t\mathbf{x}_T, g_t = g_t. \tag{56}$$

In the original paper,

$$\bar{\mathbf{f}}_t(\mathbf{X}_t) = 0, g_t^2 = \frac{d}{dt}\sigma_t^2, \bar{\mathbf{h}}_t(\mathbf{X}_t) = \frac{\mathbf{x}_T - \mathbf{x}_t}{\sigma_T^2 - \sigma_t^2}. \tag{57}$$

Therefore,

$$\bar{\mathbf{f}}_t(\mathbf{X}_t) - g_t^2\bar{\mathbf{h}}_t(\mathbf{X}_t) = \frac{2\sigma_t\dot{\sigma}_t(\mathbf{x}_T - \mathbf{x}_t)}{\sigma_T^2 - \sigma_t^2}, g_t^2 = 2\dot{\sigma}_t\sigma_t. \tag{58}$$

In our framework, $f_t, s_t, g_t^2$ can be calculated:

$$f_t = \frac{\dot{\alpha}_t}{\alpha_t} = \frac{d}{dt}\log\alpha_t = \frac{d}{dt}\log\frac{\sigma_T^2 - \sigma_t^2}{\sigma_T^2} = \frac{-2\sigma_t\dot{\sigma}_t}{\sigma_T^2 - \sigma_t^2}, \tag{59}$$

$$s_t = \dot{\beta}_t - \frac{\dot{\alpha}_t}{\alpha_t}\beta_t = \frac{2\sigma_t\dot{\sigma}_t}{\sigma_T^2} + \frac{2\sigma_t\dot{\sigma}_t}{\sigma_T^2 - \sigma_t^2}\cdot\frac{\sigma_t^2}{\sigma_T^2} = \frac{2\sigma_t\dot{\sigma}_t}{\sigma_T^2 - \sigma_t^2}. \tag{60}$$

$$g_t^2 = 2(\gamma_t\dot{\gamma}_t - \frac{\dot{\alpha}_t}{\alpha_t}\gamma_t^2) = 2\gamma_t^2\left(\frac{\dot{\gamma}_t}{\gamma_t} - \frac{\dot{\alpha}_t}{\alpha_t}\right) = \gamma_t^2\left(\frac{(\sigma_T^2 - 2\sigma_t^2)\dot{\sigma}_t}{(\sigma_T^2 - \sigma_t^2)\sigma_t} + \frac{2\dot{\sigma}_t\sigma_t}{\sigma_T^2 - \sigma_t^2}\right) = 2\sigma_t\dot{\sigma}_t. \tag{61}$$

Therefore,

$$f_t\mathbf{X}_t + s_t\mathbf{x}_T = \frac{2\sigma_t\dot{\sigma}_t(\mathbf{x}_T - \mathbf{x}_t)}{\sigma_T^2 - \sigma_t^2} = \bar{\mathbf{f}}_t(\mathbf{X}_t) - g_t^2\bar{\mathbf{h}}_t(\mathbf{X}_t), \quad g_t = g_t, \tag{62}$$

which matches the formulation in DDBM.

$\square$

## D.2  DDBM-VP

DDBM-VP can be reformulated in our framework as we set :

$$\alpha_t = a_t(1 - \frac{\sigma_t^2 a_1^2}{\sigma_1^2 a_t^2}), \beta_t = \frac{\sigma_t^2 a_1}{\sigma_1^2 a_t}, \gamma_t = \sqrt{\sigma_t^2(1 - \frac{\sigma_t^2 a_1^2}{\sigma_1^2 a_t^2})}. \tag{63}$$

*Proof.* In the original DDBM-VP setting,

$$\bar{\mathbf{f}}_t(\mathbf{X}_t) = \frac{d\log a_t}{dt}\mathbf{x}_t, \tag{64}$$

$$g_t^2 = 2\sigma_t\dot{\sigma}_t - 2\frac{\dot{a}_t}{a_t}\sigma_t^2 = \frac{2\sigma_t\dot{\sigma}_t a_t - 2\sigma_t^2\dot{a}_t}{a_t}, \tag{65}$$

$$\bar{\mathbf{h}}_t(\mathbf{X}_t) = \frac{(a_t/a_1)\mathbf{x}_T - \mathbf{x}_t}{\sigma_t^2(\text{SNR}_t/\text{SNR}_1 - 1)} = \frac{a_1 a_t\mathbf{x}_T - a_1^2\mathbf{x}_t}{\sigma_1^2 a_t^2 - \sigma_t^2 a_1^2}. \tag{66}$$

Therefore,

$$\bar{\mathbf{f}}_t(\mathbf{X}_t) - g_t^2 \bar{\mathbf{h}}_t(\mathbf{X}_t) = \left[ \frac{\dot{a}_t}{a_t} - \frac{2\sigma_t a_1^2(\dot{\sigma}_t a_t - \sigma_t \dot{a}_t)}{a_t(\sigma_1^2 a_t^2 - \sigma_t^2 a_1^2)} \right] \mathbf{x}_t + \frac{2\sigma_t a_1(\dot{\sigma}_t a_t - \sigma_t \dot{a}_t)}{\sigma_1^2 a_t^2 - \sigma_t^2 a_1^2} \mathbf{x}_T. \tag{67}$$

In our framework, $f_t, s_t, g_t^2$ can be calculated:

$$f_t = \frac{\dot{\alpha}_t}{\alpha_t} = \frac{d}{dt} \log \alpha_t \tag{68}$$

$$= \frac{d}{dt} \log \frac{\sigma_1^2 a_t^2 - \sigma_t^2 a_1^2}{\sigma_1^2 a_t} \tag{69}$$

$$= \frac{2\sigma_1^2 a_t \dot{a}_t - 2a_1^2 \sigma_t \dot{\sigma}_t}{\sigma_1^2 a_t^2 - \sigma_t^2 a_1^2} - \frac{\dot{a}_t}{a_t} \tag{70}$$

$$= \frac{\dot{a}_t}{a_t} - \frac{2a_1^2 \sigma_t(a_t \dot{\sigma}_t - \dot{a}_t \sigma_t)}{a_t(\sigma_1^2 a_t^2 - \sigma_t^2 a_1^2)}, \tag{71}$$

$$s_t = \dot{\beta}_t - \frac{\dot{\alpha}_t}{\alpha_t} \beta_t = \beta_t \left( \frac{\dot{\beta}_t}{\beta_t} - \frac{\dot{\alpha}_t}{\alpha_t} \right) \tag{72}$$

$$= \frac{\sigma_t^2 a_1}{\sigma_1^2 a_t} \left( \frac{2\dot{\sigma}_t}{\sigma_t} - \frac{2\sigma_1^2 a_t \dot{a}_t - 2a_1^2 \sigma_t \dot{\sigma}_t}{\sigma_1^2 a_t^2 - \sigma_t^2 a_1^2} \right) \tag{73}$$

$$= \frac{2\sigma_t a_1(\dot{\sigma}_t a_t - \sigma_t \dot{a}_t)}{\sigma_1^2 a_t^2 - \sigma_t^2 a_1^2}, \tag{74}$$

$$g_t^2 = \gamma_t \dot{\gamma}_t - \frac{\dot{\alpha}_t}{\alpha_t} \gamma_t^2 = \gamma_t^2 \left( \frac{\dot{\gamma}_t}{\gamma_t} - \frac{\dot{\alpha}_t}{\alpha_t} \right) \tag{75}$$

$$= \gamma^2 \frac{d}{dt} \log \frac{\gamma_t}{\alpha_t} \tag{76}$$

$$= \gamma^2 \frac{d}{dt} \left( \frac{1}{2} \log \frac{\sigma_t^2 \sigma_1^2}{\sigma_1^2 a_t^2 - \sigma_t^2 a_1^2} \right) \tag{77}$$

$$= \sigma_t^2 \left( 1 - \frac{\sigma_t^2 a_1^2}{\sigma_1^2 a_t^2} \right) \left( \frac{\dot{\sigma}_t}{\sigma_t} - \frac{\sigma_1^2 a_t \dot{a}_t - a_1^2 \sigma_t \dot{\sigma}_t}{\sigma_1^2 a_t^2 - \sigma_t^2 a_1^2} \right) \tag{78}$$

$$= \frac{\dot{\sigma}_t \sigma_t a_t - \sigma_t^2 \dot{a}_t}{a_t}. \tag{79}$$

Therefore,

$$f_t \mathbf{X}_t + s_t \mathbf{x}_T == \bar{\mathbf{f}}_t(\mathbf{X}_t) - g_t^2 \bar{\mathbf{h}}_t(\mathbf{X}_t), g_t = g_t, \tag{80}$$

which matches the formulation in DDBM.

$\square$

### D.3 EDM

**ODE formulation**. The ODE formulation in EDM can be formlated in our framework as we set $\alpha_t = 1, \beta_t = 0, \gamma_t = \sigma_t$.

*Proof.* Recall 25, the ODE formulation is given by:

$$d\mathbf{X}_t = \left[ f_t \mathbf{X}_t + s_t \mathbf{x}_T - \frac{1}{2} g_t^2 \nabla_{\mathbf{X}_t} \log p_t(\mathbf{X}_t | \mathbf{x}_T) \right] dt \quad \mathbf{X}_T = \mathbf{x}_T \tag{81}$$

where $f_t = \frac{\dot{\alpha}_t}{\alpha_t}$, $s_t = \dot{\beta}_t - \frac{\dot{\alpha}_t}{\alpha_t}\beta_t$, $g_t = \sqrt{2(\gamma_t\dot{\gamma}_t - \frac{\dot{\alpha}_t}{\alpha_t}\gamma_t^2)}$. As $\alpha_t = 1, \beta_t = 0, \gamma_t = \sigma_t$, The sampling ODE is given by:

$$dX_t = -\sigma_t\dot{\sigma}_t\nabla_{\mathbf{x}_t}\log p_t(X_t)dt \tag{82}$$

$\square$

**Sampling SDEs with noise added**. Recall Proposition 4.1, as $\alpha_t = 1, \beta_t = 0, \gamma_t = \sigma_t$, then the SDE has the form:

$$dX_t = (-\sigma_t\dot{\sigma}_t + \epsilon_t)\nabla_{\mathbf{x}_t}\log p_t(X_t)dt + \sqrt{2\epsilon_t}dW_t. \tag{83}$$

Now we recover the stochastic sampling SDE in original EDM paper.

### D.4   I2SB

I2SB can be reformulated in our framework as we let:

$$\alpha_t = 1 - \frac{\sigma_t^2}{\sigma_1^2}, \beta_t = \frac{\sigma_t^2}{\sigma_1^2}, \gamma_t = \sqrt{\sigma_t^2(1 - \frac{\sigma_t^2}{\sigma_1^2})} \tag{84}$$

where $\sigma_t^2 := \int_0^t \beta_\tau d\tau$.

When $2\epsilon_t\Delta t = \gamma_{t-\Delta t}^2 - \beta_{t-\Delta t}^2\gamma_t^2/\beta_t^2$, the coefficient of $x_T$ in Eq. (12) vanishes. Thus, Eq. (12) can be simplified as:

$$x_{t-\Delta t} = (\alpha_{t-\Delta t} - \alpha_t\frac{\beta_{t-\Delta t}}{\beta_t})\hat{x}_0 + \frac{\beta_{t-\Delta t}}{\beta_t}x_t + \sqrt{\gamma_{t-\Delta t}^2 - \frac{\beta_{t-\Delta t}^2\gamma_t^2}{\beta_t^2}}\bar{z}_t \tag{85}$$

Using discretization in Eq. (85):

$$\mathbf{x}_{t-\Delta t} = (\alpha_{t-\Delta t} - \alpha_t\frac{\beta_{t-\Delta t}}{\beta_t})\hat{\mathbf{x}}_0 + \frac{\beta_{t-\Delta t}}{\beta_t}\mathbf{x}_t + \sqrt{\gamma_{t-\Delta t}^2 - \frac{\beta_{t-\Delta t}^2\gamma_t^2}{\beta_t^2}}\bar{\mathbf{z}}_t \tag{86}$$

$$= (1 - \frac{\beta_{t-\Delta t}}{\beta_t})\hat{\mathbf{x}}_0 + \frac{\beta_{t-\Delta t}}{\beta_t}\mathbf{x}_t + \sqrt{\gamma_{t-\Delta t}^2 - \frac{\beta_{t-\Delta t}^2\gamma_t^2}{\beta_t^2}}\bar{\mathbf{z}}_t \tag{87}$$

$$= (1 - \frac{\sigma_{t-\Delta t}^2}{\sigma_t^2})\hat{\mathbf{x}}_0 + \frac{\sigma_{t-\Delta t}^2}{\sigma_t^2}\mathbf{x}_t + \sqrt{\frac{\sigma_{t-\Delta t}^2(1 - \frac{\sigma_{t-\Delta t}^2}{\sigma_1^2})\frac{\sigma_t^4}{\sigma_1^4} - \frac{\sigma_{t-\Delta t}^4}{\sigma_1^4}\sigma_t^2(1 - \frac{\sigma_t^2}{\sigma_1^2})}{\frac{\sigma_t^4}{\sigma_1^4}}}\bar{\mathbf{z}}_t \tag{88}$$

$$= (1 - \frac{\sigma_{t-\Delta t}^2}{\sigma_t^2})\hat{\mathbf{x}}_0 + \frac{\sigma_{t-\Delta t}^2}{\sigma_t^2}\mathbf{x}_t + \sqrt{\frac{\sigma_{t-\Delta t}^2(\sigma_t^2 - \sigma_{t-\Delta t}^2)}{\sigma_t^2}}\bar{\mathbf{z}}_t \tag{89}$$

In the I2SB paper, define $a_n^2 := \int_{t_n}^{t_{n+1}}\beta_\tau d\tau$, $\sigma_n^2 := \int_0^{t_n}\beta_\tau d\tau$. Therefore,

$$\mathbf{x}_n = \frac{a_n^2}{a_n^2 + \sigma_n^2}\hat{\mathbf{x}}_0 + \frac{\sigma_n^2}{a_n^2 + \sigma_n^2}\mathbf{x}_{n+1} + \sqrt{\frac{\sigma_n^2 a_n^2}{\alpha_n^2 + \sigma_n^2}}\bar{\mathbf{z}}_t \tag{90}$$

Thus, we reproduce the sampler of I2SB.

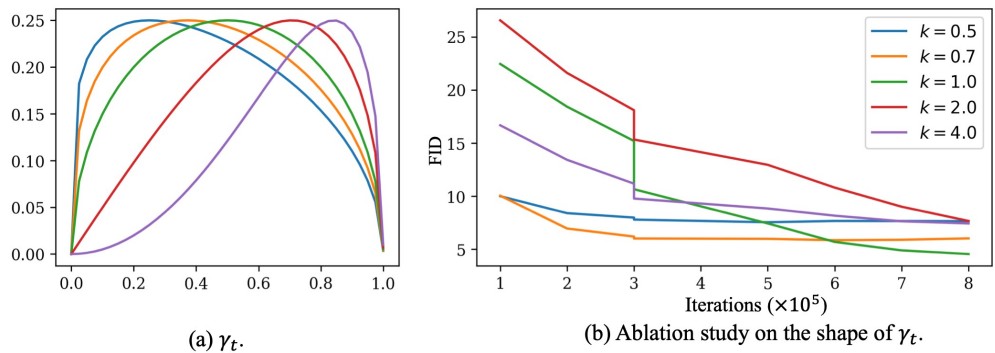

(a) $\gamma_t$.  (b) Ablation study on the shape of $\gamma_t$.

Figure 7: Ablation study on the shape of $\gamma_t$.

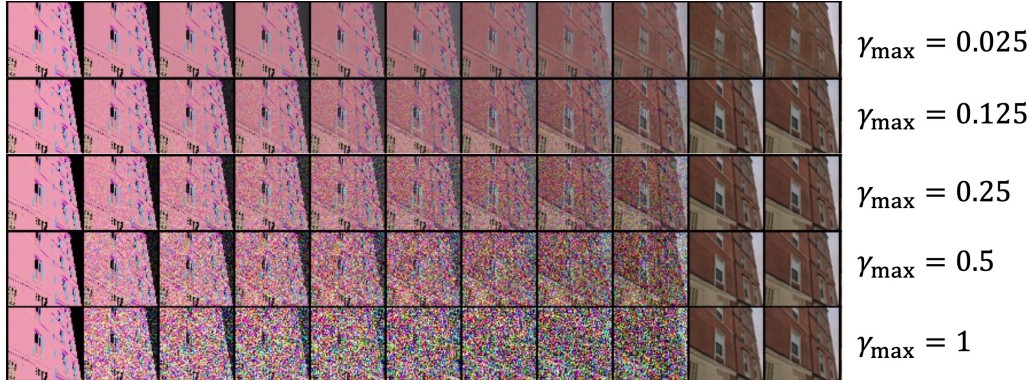

Figure 8: Sampling paths with dfferent choices of $\gamma_t$. As $\gamma_t$ extreamly low, e.g, $\gamma_{\max} = 0.025$, the model will be failed to construct details of images.

# E  Additional design guideline

$\alpha_t$ **and** $\beta_t$**.** Theoretically, $\alpha_t$ and $\beta_t$ can be freely designed, and future work may explore alternative design choices. However, in this paper, we focus on the simple case where $\alpha_t = 1-t$ and $\beta_t = t$. The rationale is as follows: consider the scenario where $\alpha_t = 1 - \beta_t$, which represents an interpolation along the line segment between $x_0$ and $x_1$. For the path $p_t^{(1)}(x) = \mathcal{N}((1-\beta_t)x_0 + \beta_t x_1, \gamma_t^2 \mathbb{I})$, where $\beta_t$ is invertible, it is straightforward to construct another path $p_t^{(2)}(x) = \mathcal{N}((1-t)x_0 + tx_1, \gamma_{\beta_t^{-1}}^2 \mathbb{I})$, which achieves the same objective function but uses a different distribution of $t$ during training. Based on this equivalence, setting $\alpha_t = 1 - t$ and $\beta_t = t$ is a reasonable choice.

**The shape of** $\gamma_t$**.** We conducted an ablation study on $\gamma_t$ with different shapes. Specifically, we assumed $\gamma_t$ has the form $\gamma_t = 2\gamma_{\max}\sqrt{t^k(1-t^k)}$, as shown in Fig. 7, $\gamma_t$ will have different shape as we set different $k$. The results indicate that the best performance is achieved when $k = 1$, which is the exact setting used in this paper.

$\gamma_{\max}$. Our ablation studies on $\gamma_{\max}$ demonstrate that the optimal values of $\gamma_{\max}$ are approximately $0.125$ or $0.25$. Furthermore, the sampling paths corresponding to different choices of $\gamma_t$ are shown in Fig. 8. Adding an appropriate amount of noise to the transition kernel helps in constructing finer details.

$\epsilon_t$. We use the setting $\epsilon_t = \eta\left(\gamma_t\dot{\gamma}_t - \frac{\dot{\alpha}_t}{\alpha_t}\gamma_t^2\right)$. The ablation studies on $\epsilon_t$ demonstrate that the optimal choice of $\eta$ for the DDBM-VP model is approximately $0.3$, while the best choice for the ECSI model with a Linear Path is around $1.0$. Additionally, we present sample paths and generated images under different $\eta$ settings to illustrate heuristic parameter tuning techniques. The results are shown in Figures 10, 11, and 12. Too small a value of $\eta$ results in the loss of high-frequency information, while too large a value of $\eta$ produces over-sharpened and potentially noisy sampled images.

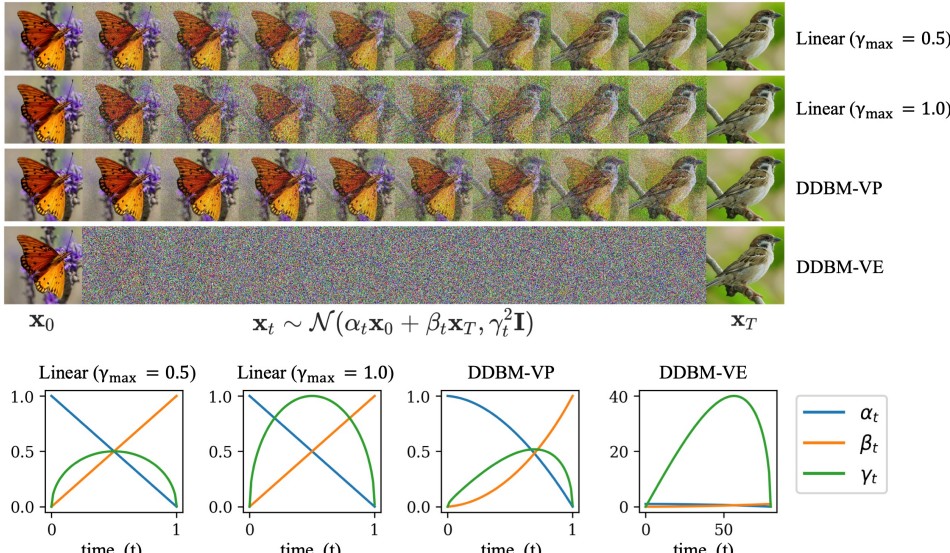

Figure 9: An illustration of design choices of transition kernels and how they affect the I2I translation process. $\alpha_t$ and $\beta_t$ define the interpolation between two images, while $\gamma_t$ controls the noise added to the process. ntuitively, the DDBM-VE model introduces excessive noise in the middle stages, which is unnecessary for effective image translation and may explain its poor performance. In contrast, our Linear path results in a symmetrical noise schedule, ensuring a more balanced process. On the other hand, the DDBM-VP path adds more noise near $\mathbf{x}_T$, , indicating that during training, more computational resources are focused around $\mathbf{x}_0$.

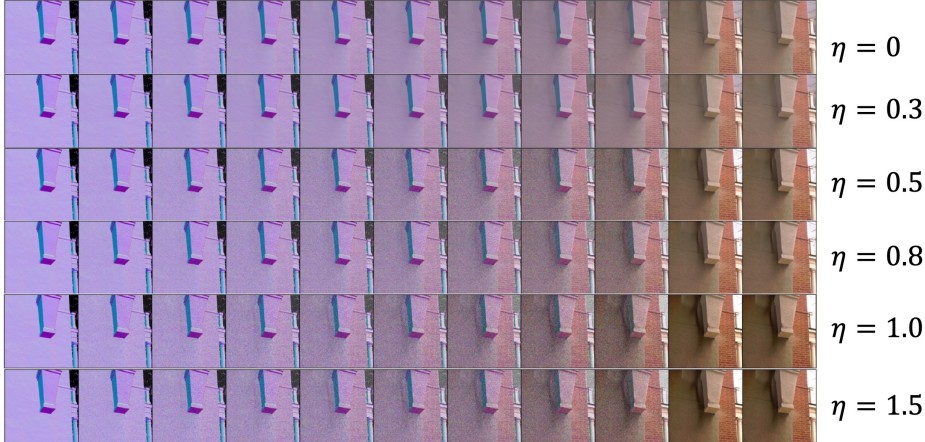

Figure 10: Sampling path with dfferent choices of $\epsilon_t$. As $\epsilon_t = 0$, the generated images lack details, as $\epsilon_t$ too large, the sampled images are over-sharpening. The best choices of $\epsilon_t$ are around $\epsilon_t = 0.8$ and $\epsilon_t = 1.0$.

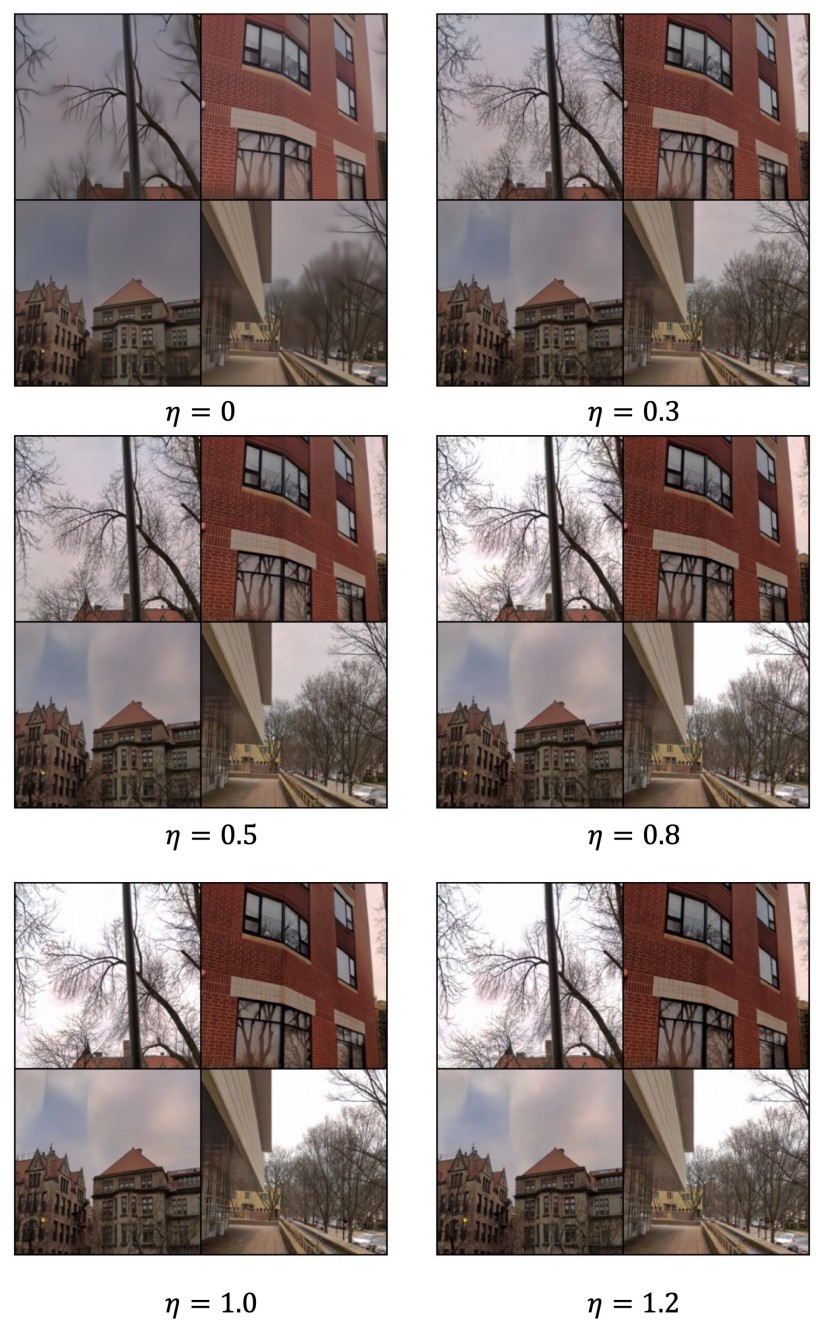

$\eta = 0$             $\eta = 0.3$

$\eta = 0.5$           $\eta = 0.8$

$\eta = 1.0$           $\eta = 1.2$

Figure 11: Comparison of sampled images with different $\epsilon_t$ for ECSI model, where $\epsilon_t = \eta(\gamma_t \dot{\gamma}_t - \frac{\dot{\alpha}_t}{\alpha_t}\gamma_t^2)$, $\gamma_{\max} = 0.25$, $b = 0$.

## F  Impact Statement

Our method can improve image translation and solving inverse problem, which may benefit applications in medical imaging. However, it is important to note that as with many generative and restoration models, our method could be misused for malicious image manipulation.

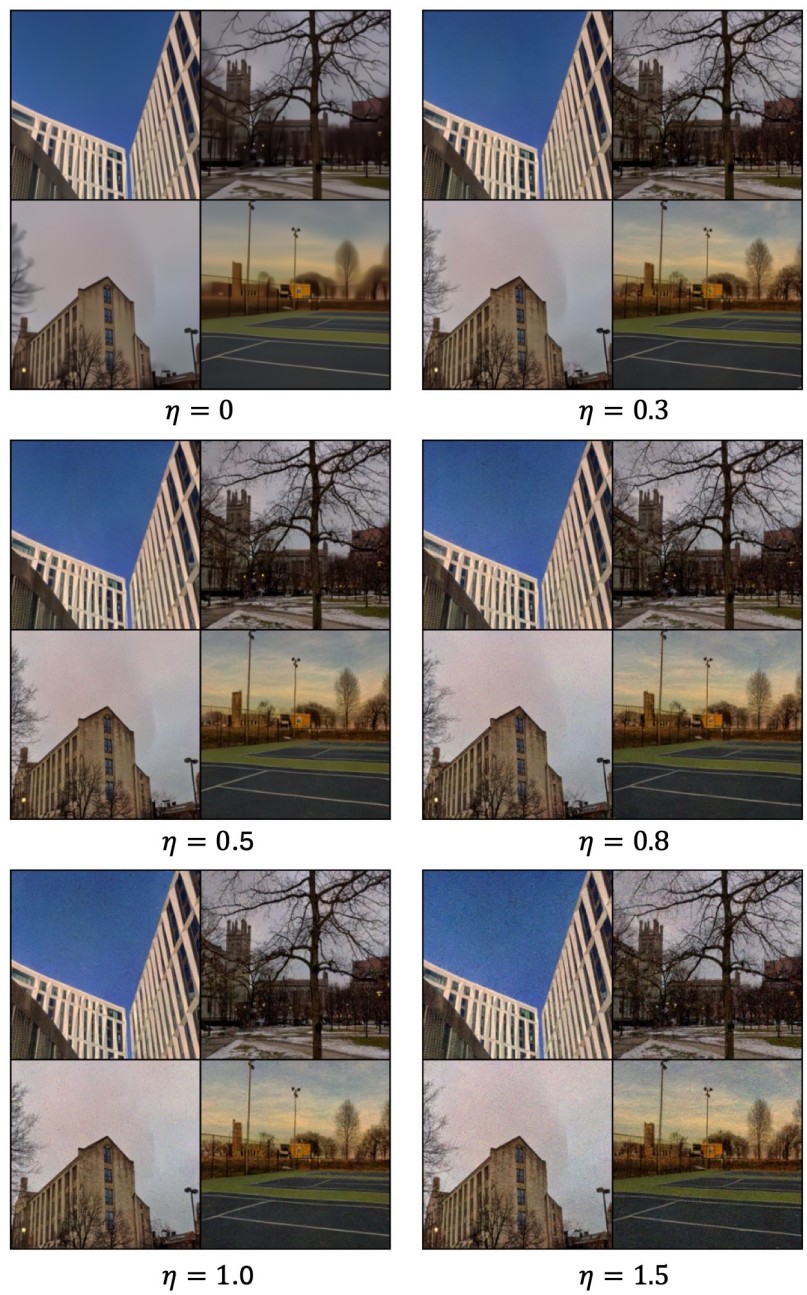

Figure 12: Comparison of sampled images with different $\epsilon_t$ for DDBM-VP pretrained model, where $\epsilon_t = \eta(\gamma_t\dot{\gamma}_t - \frac{\dot{\alpha}_t}{\alpha_t}\gamma_t^2)$.

## G   Experiment Details

**Architecture**. We maintain the architecture and parameter settings consistent with [18], utilizing the ADM model [10] for $64 \times 64$ resolution, modifying the channel dimensions from 192 to 256 and reducing the number of residual blocks from three to two. Apart from these changes, all other settings remain identical to those used for $64 \times 64$ resolution.

**Training**. We include additional pre- and post-processing steps: scaling functions and loss weighting, the same ingredient as [17]. Let $D_\theta(\mathbf{x}_t, \mathbf{x}_T, t) = c_{\text{skip}}(t)\mathbf{x}_t + c_{\text{out}(t)}(t)F_\theta(c_{\text{in}}(t)\mathbf{x}_t, c_{\text{noise}}(t))$,

where $F_\theta$ is a neural network with parameter $\theta$, the effective training target with respect to the raw network $F_\theta$ is: $\mathbb{E}_{\mathbf{x}_t, \mathbf{x}_0, \mathbf{x}_T, t}\left[\lambda \| c_{\text{skip}}(\mathbf{x}_t + c_{\text{out}} F_\theta(c_{\text{in}} \mathbf{x}_t, c_{\text{noise}}) - \mathbf{x}_0 \|^2\right]$. Scaling scheme are chosen by requiring network inputs and training targets to have unit variance $(c_{\text{in}}, c_{\text{out}})$, and amplifying errors in $F_\theta$ as little as possible. Following reasoning in [18],

$$c_{\text{in}}(t) = \frac{1}{\sqrt{\alpha_t^2 \sigma_0^2 + \beta_t^2 \sigma_T^2 + 2\alpha_t \beta_t \sigma_{0T} + \gamma_t^2}}, \quad c_{\text{skip}}(t) = (\alpha_t \sigma_0^2 + \beta_t \sigma_{0T}) * c_{\text{in}}^2, \quad (91)$$

$$c_{\text{out}}(t) = \sqrt{\beta_t^2 \sigma_0^2 \sigma_1^2 - \beta_t^2 \sigma_{0T}^2 + \gamma_t^2 \sigma_0^2} c_{\text{in}}, \quad \lambda = \frac{1}{c_{\text{out}}^2}, \quad c_{\text{noise}}(t) = \frac{1}{4}\log(t), \quad (92)$$

where $\sigma_0^2, \sigma_T^2$, and $\sigma_{0T}$ denote the variance of $\mathbf{x}_0$, variance of $\mathbf{x}_T$ and the covariance of the two, respectively.

We note that TrigFlow [23], adopts the same score reparameterization and pre-conditioning techniques. It can be considered a special case of our framework by setting $\alpha_t = \cos(t)$, $\beta_t = 0$, $\gamma_t = \sigma_0 \sin(t)$, $t \in [0, \frac{\pi}{2}]$. In this case, $\sigma_T = 0$, $\sigma_{0T} = 0$,

$$c_{\text{in}}(t) = \frac{1}{\sqrt{\alpha_t^2 \sigma_0^2 + \gamma_t^2}} = \frac{1}{\sqrt{\sin^2(t)\sigma_0^2 + \cos^2(t)\sigma_0^2}} = \frac{1}{\sigma_0}, \quad (93)$$

$$c_{\text{skip}}(t) = (\alpha_t \sigma_0^2) c_{in}^2 = \cos(t) \cdot \sigma_0^2 \cdot \frac{1}{\sigma_0^2} = \cos(t), \quad (94)$$

$$c_{out}(t) = \sqrt{\gamma_t^2 \sigma_0^2} \cdot c_{in} = \sin(t)\sigma_0, \quad (95)$$

$$D_\theta(x_t, t) = c_{\text{skip}} x_t + c_{\text{out}} F_\theta(c_{\text{in}} x_t, c_{\text{noise}}) = \cos(t) x_t + \sin(t)\sigma_0 F_\theta(\frac{1}{\sigma_0}, c_{\text{noise}}). \quad (96)$$

Then we recover TrigFlow.

In our implementation, we set $\sigma_0 = \sigma_T = 0.5$, $\sigma_{0T} = \sigma_0^2/2$ for all training sessions. Other setting are shown in Table 9.

Table 9: Training settings

| | Dataset | edges→handbags | edges→handbags | edges→handbags |
|---|---|---|---|---|
| Model | $\eta$ | 0 | 0 | 0.5 |
| | $\gamma_{\max}$ | 0.125 | 0.25 | 0.125 |
| | GPU | 1 A6000 48G | 1 H100 96G | 1 H100 96G |
| | Batch size | 32 | 128 | 200 |
| Setting | Learning rate | $1 \times 10^{-5}$ | $5 \times 10^{-5}$ | $1 \times 10^{-4}$ |
| | epochs | 2078 | 2106 | 1443 |
| | Training time | 42 days | 8 days | 11 days |
| | Dataset | DIODE ($256 \times 256$) | DOIDE ($256 \times 256$) | |
| Model | $\eta$ | 0 | 0 | |
| | $\gamma_{\max}$ | 0.125 | 0.25 | |
| | GPU | 1 H100 96G | 1 H100 96G | |
| | Batch size | 16 | 16 | |
| Setting | Learning rate | $2 \times 10^{-5}$ | $2 \times 10^{-5}$ | |
| | epochs | 2617 | 1745 | |
| | Training time | 17 days | 25 days | |

**Sampling**. We use the same timesteps distributed according to EDM [17]: $(t_{\max}^{1/\rho} + \frac{i}{N}(t_{\min}^{1/\rho} - t_{\max}^{1/\rho}))^\rho$, where $t_{\min} = 0.001$ and $t_{\max} = 1 - 10^{-4}$. The best performance achieved by setting $\rho = 0.6$ for Edges2handbags and $\rho = 0.8$ for DIODE datasets.

# H  Licenses

- Edges→Handbags [16]: BSD license.
- DIODE-Outdoor [37]: MIT license.

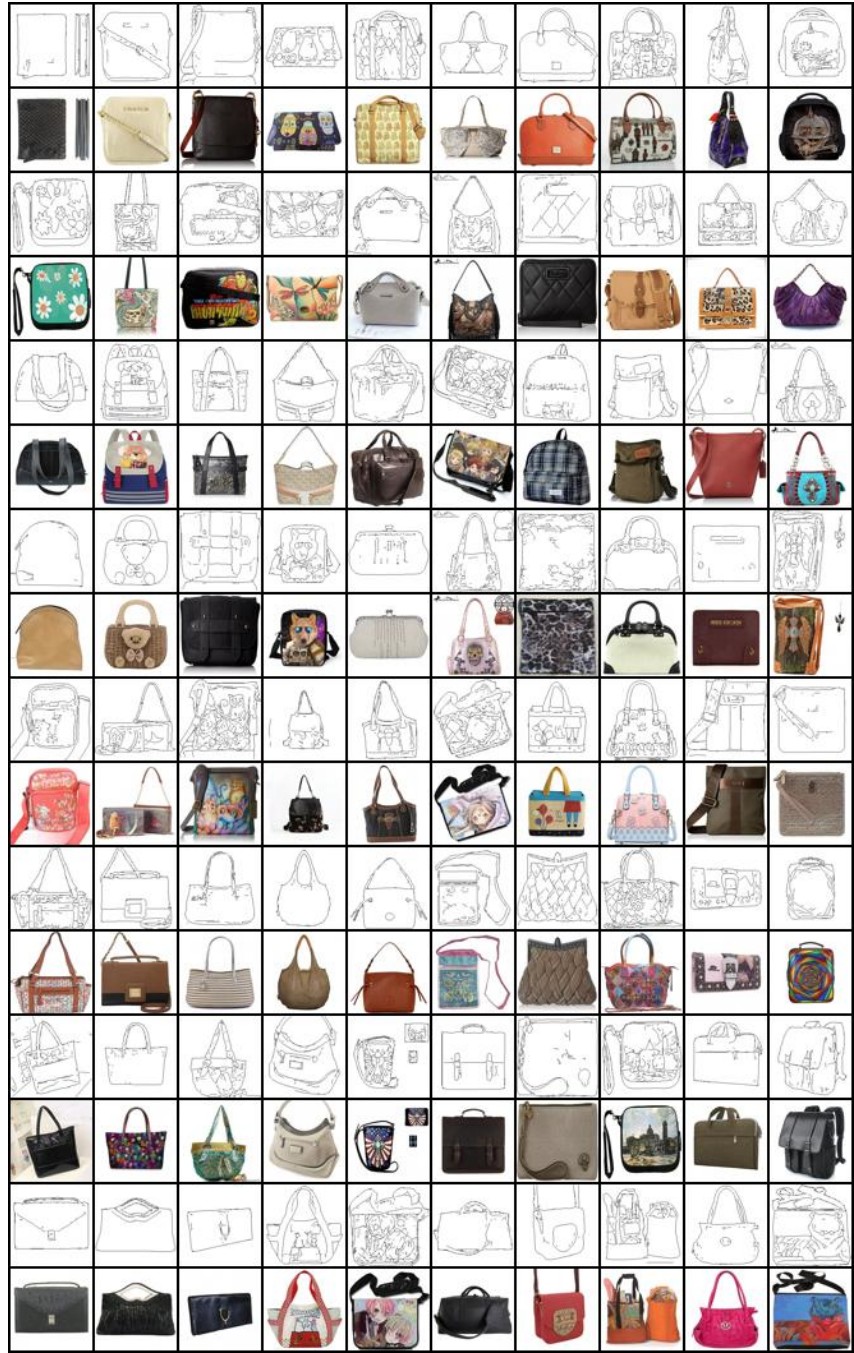

Figure 13: ECSI model and sampler ( $\gamma_{\max} = 0.125$, $\eta = 1$, $b = 0$, NFE=5, FID=0.89).

# I Additional visualizations

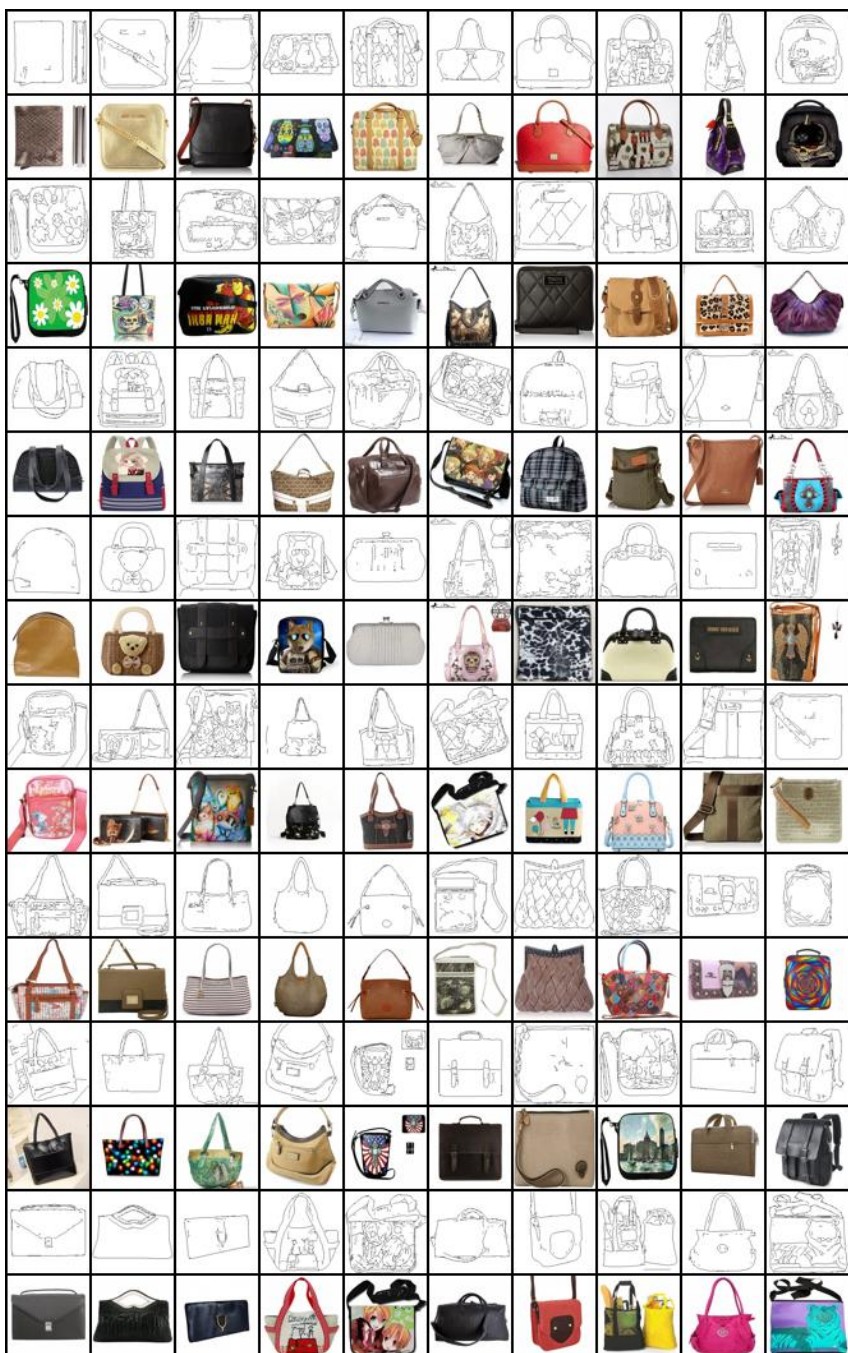

Figure 14: DDBM model and Our sampler (NFE=20, FID=1.53).

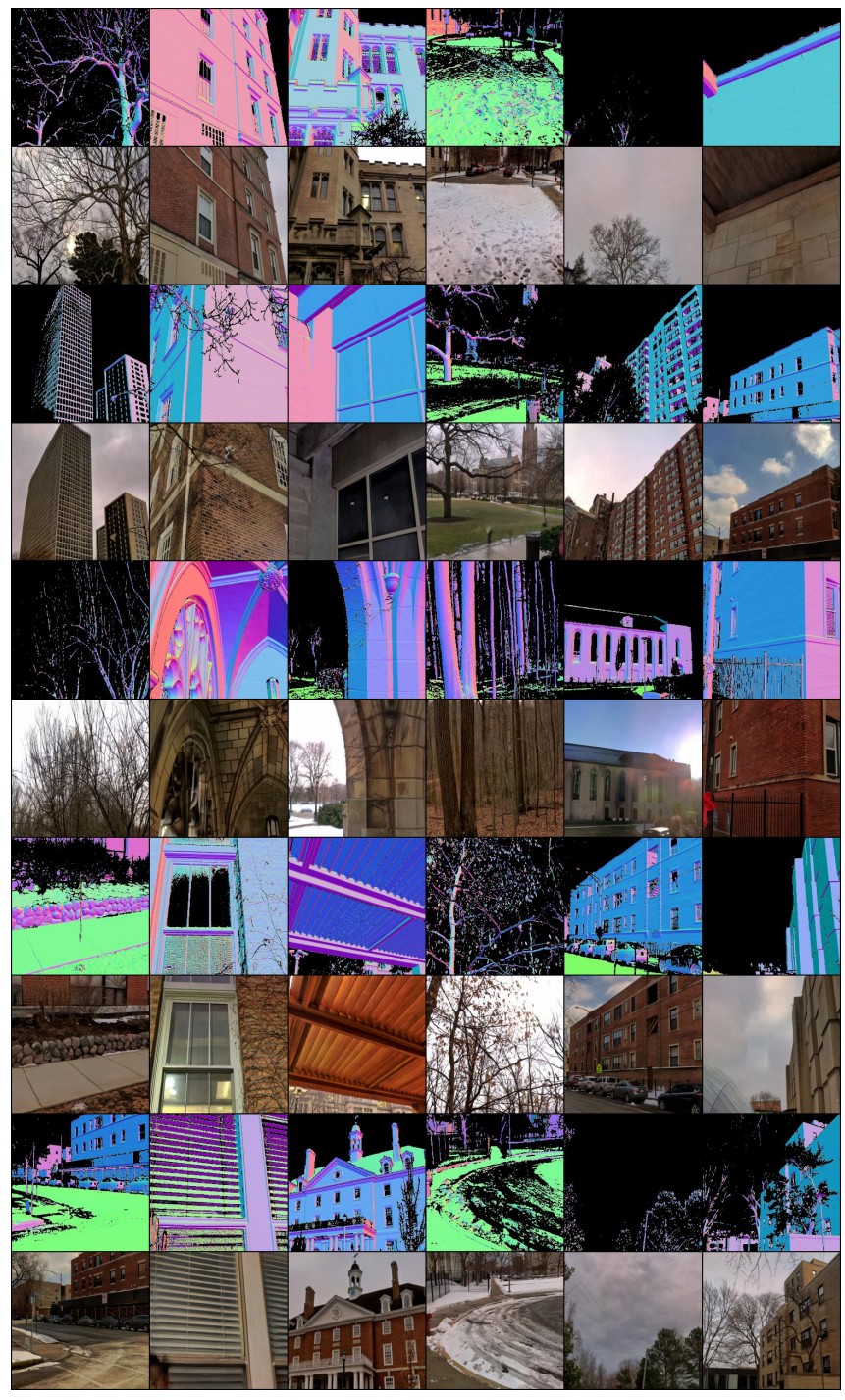

Figure 15: DDBM model and ECSI sampler ($\eta = 0.3$, NFE=20, FID=4.12). Samples for DIODE dataset (conditoned on depth images).

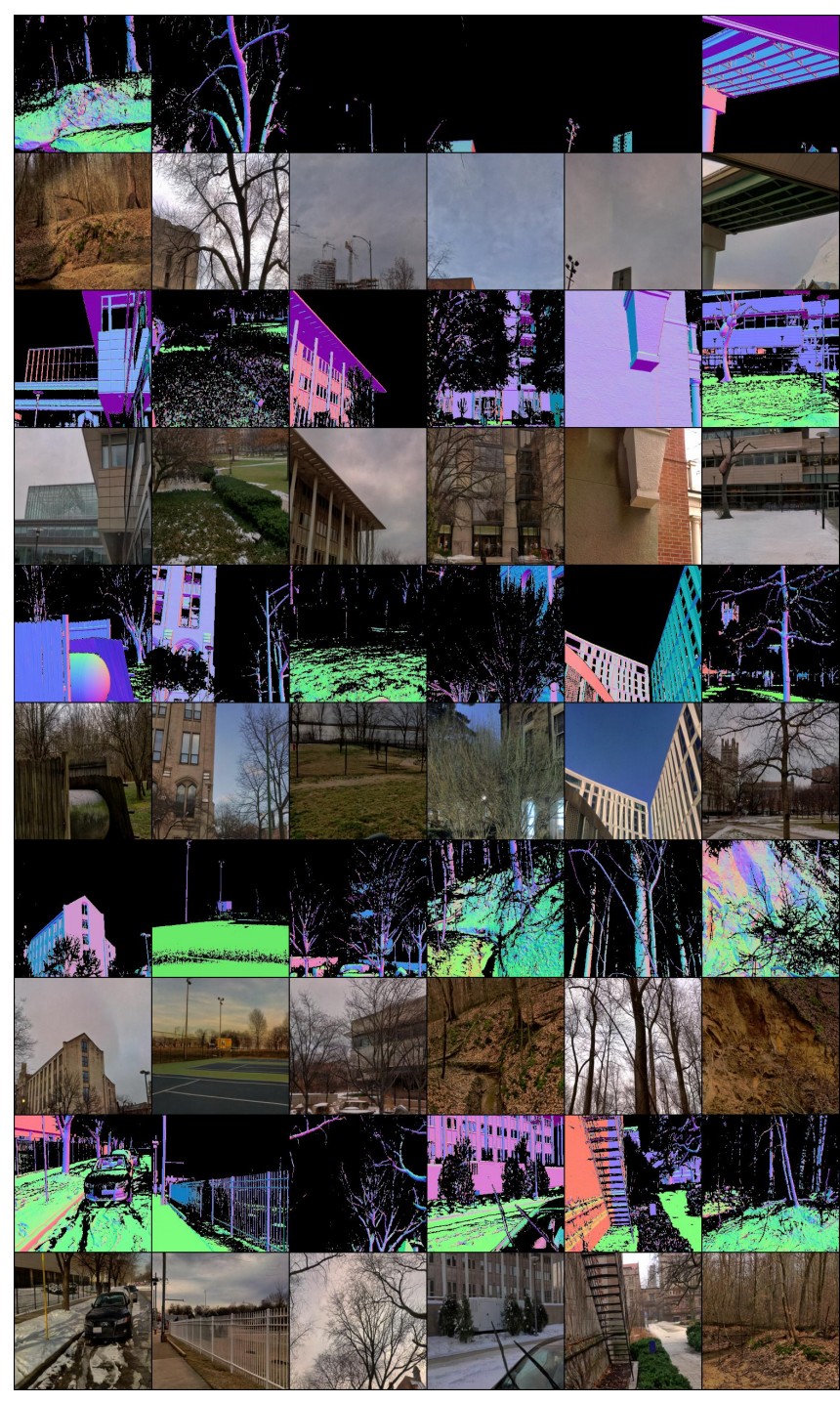

Figure 16: ECSI model and sampler ($\gamma_{\max} = 0.25$, $\eta = 1.0$, $b = 0$, NFE=5, FID = 4.16).

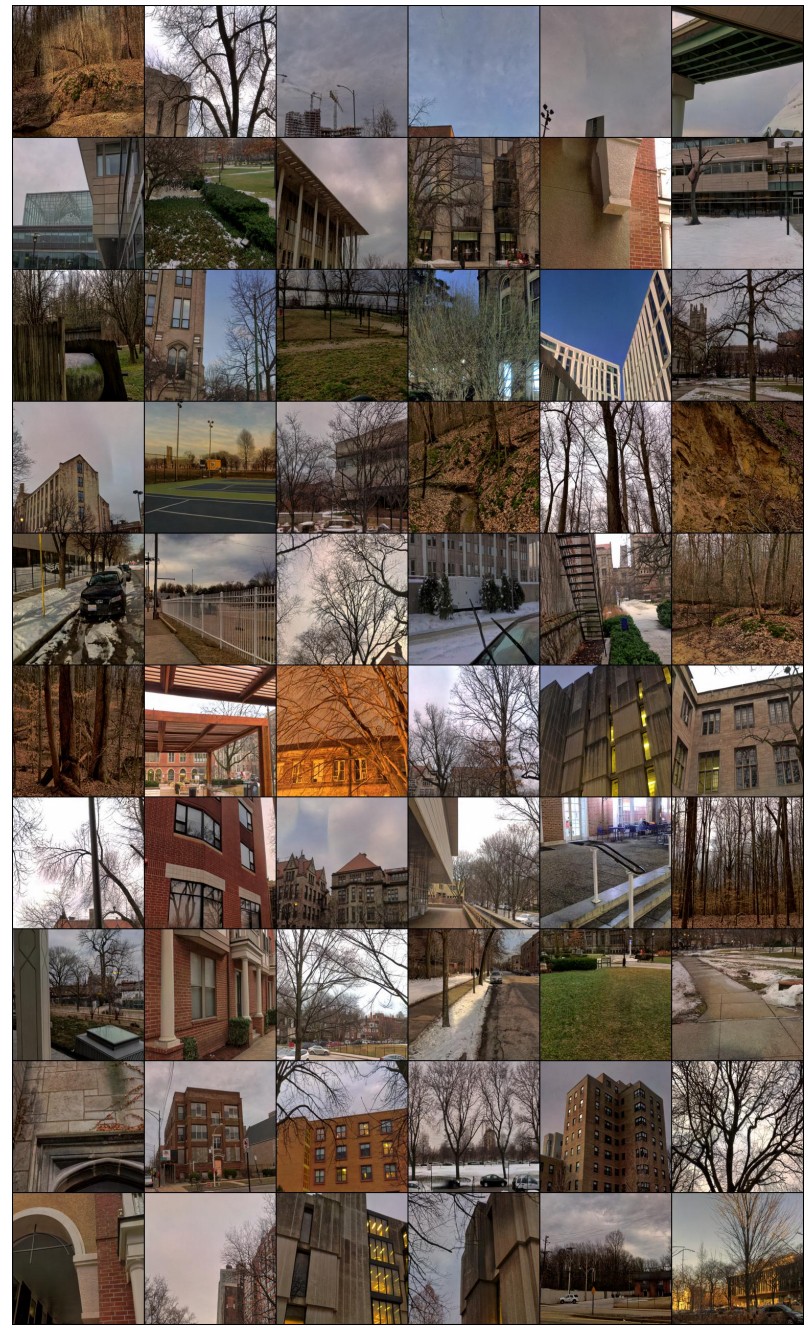

Figure 17: ECSI model and sampler ($\gamma_{\max} = 0.25$, $\eta = 1.0$, $b = 0$, NFE=20, FID = 3.27).

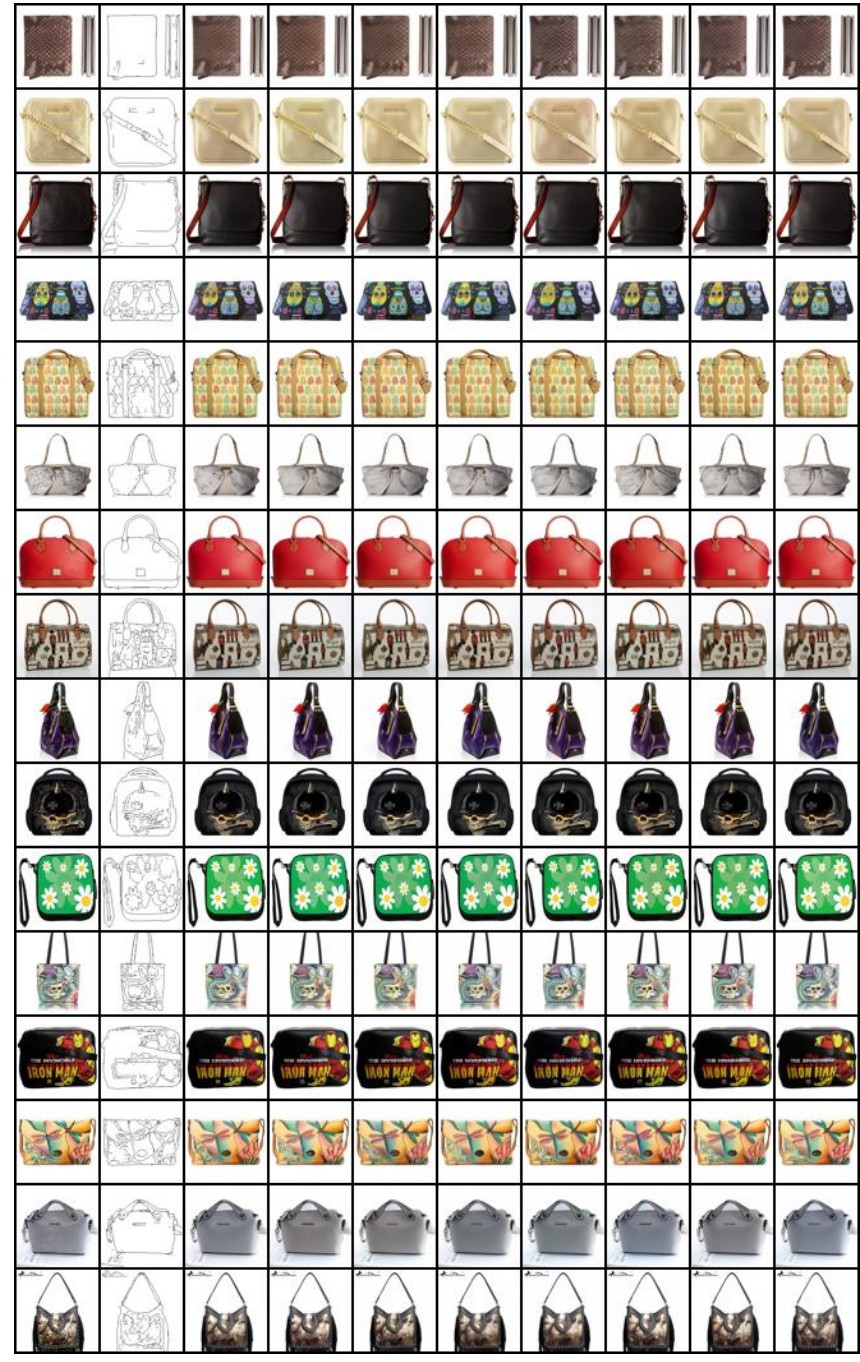

Figure 18: DDBM model and DBIM sampler (NFE=10, FID = 2.46, AFD=5.20).

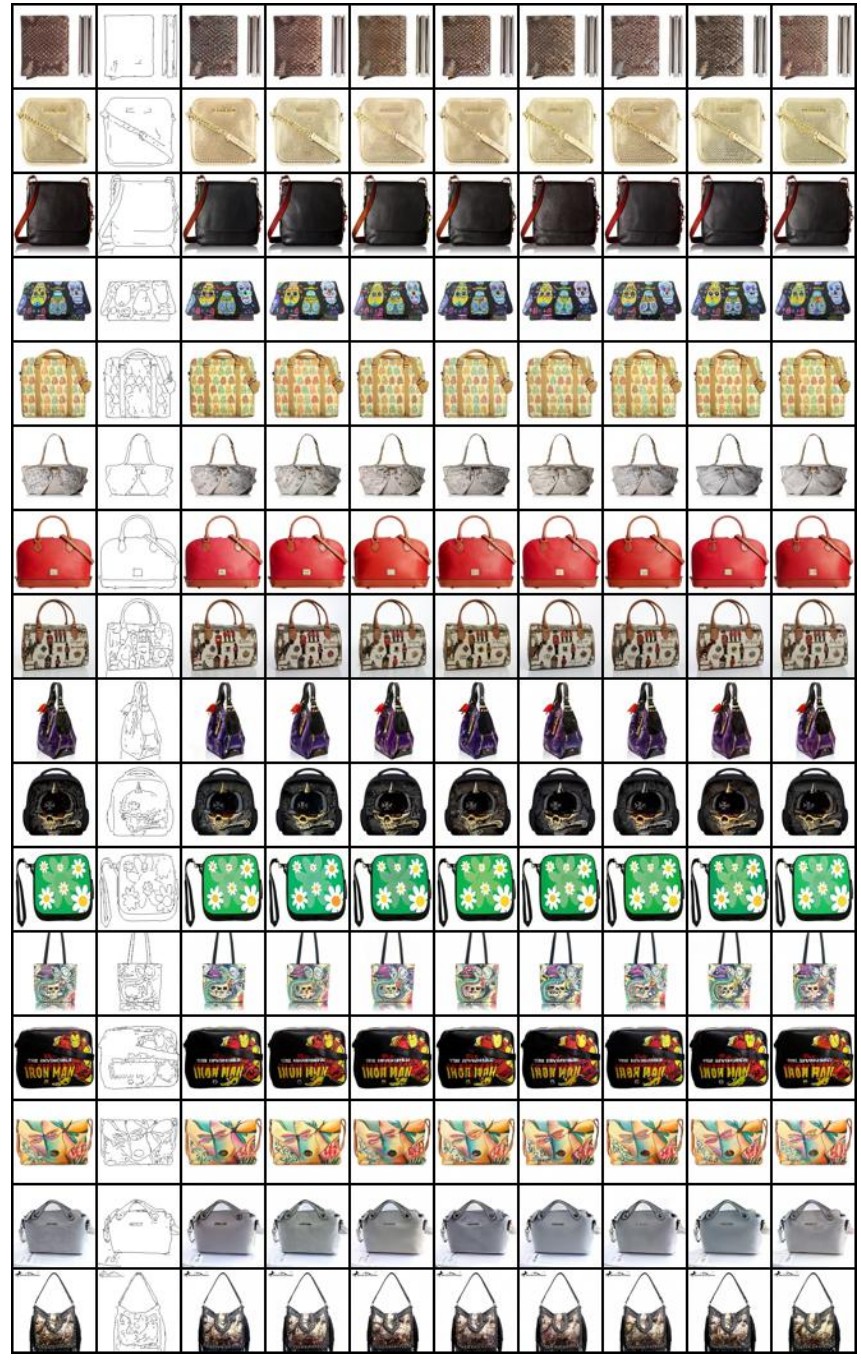

Figure 19: DDBM model and sampler (NFE=118, FID = 1.83, AFD=6.99).

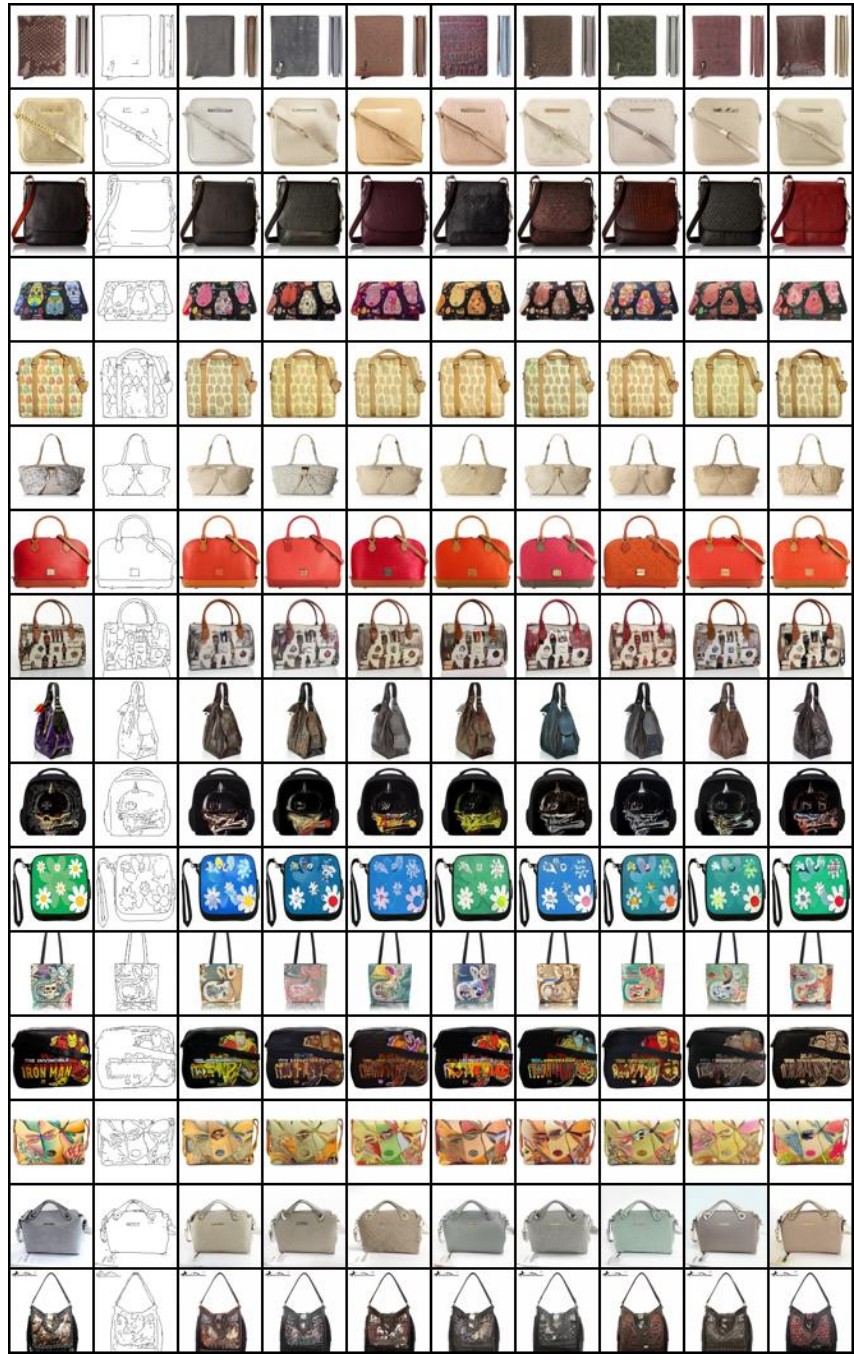

Figure 20: ECSI model and sampler ($\gamma_{\max} = 0.125$, $b = 1.0$, NFE=10, FID = 2.07, AFD=9.35).

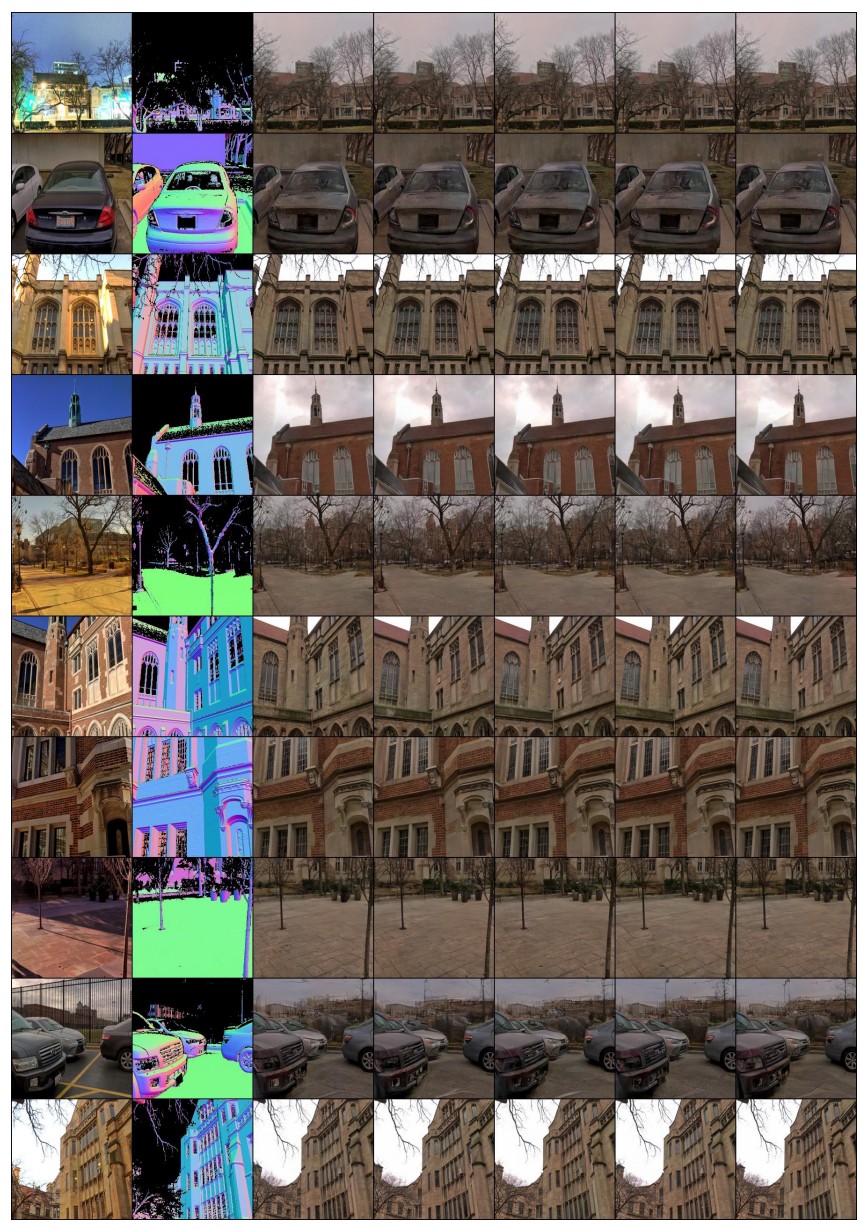

Figure 21: DDBM model and ECSI sampler on $446$ test images. (NFE=20, FID = 52.01, AFD=5.60).

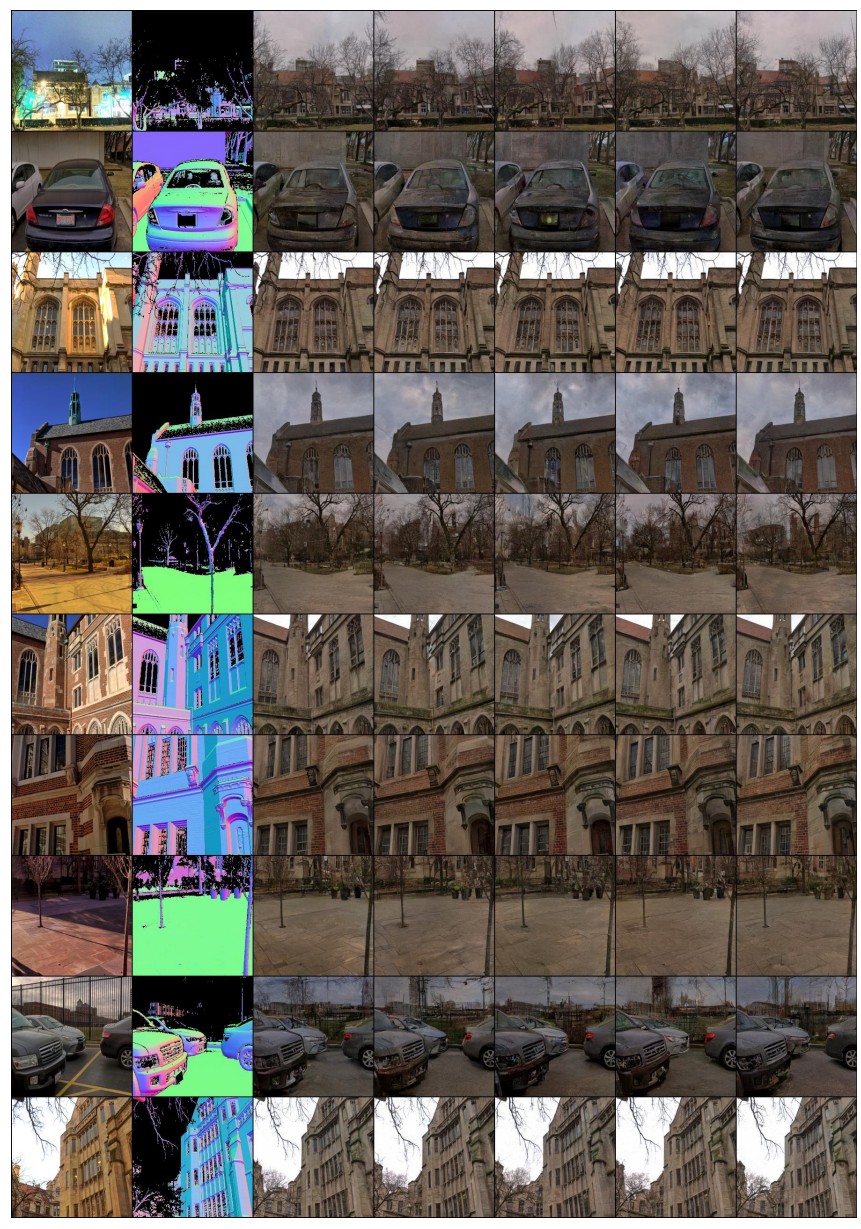

Figure 22: ECSI model and sampler on $446$ test images. ($\gamma_{\max} = 0.125$, $b = 0.5$, NFE=20, FID = 55.93, AFD=7.39).

