# OpenReview forum: "Exploring the Design Space of Diffusion Bridge Models"
_NeurIPS.cc/2025/Conference — NeurIPS 2025 poster_

### Official Review · Reviewer_CniD · 2025-06-29

**Clarity:** 2
**Significance:** 2
**Originality:** 2
**Rating:** 4
**Confidence:** 4

**Summary:**

This paper proposed a flexible framework for diffusion bridge models used in image-to-image translation. It extends existing stochastic interpolant methods by incorporating endpoint conditioning, preconditioning, and a tunable noise-based sampling strategy, thereby enlarging the design space of both bridge paths and samplers.

Additionally, the authors identify a key limitation in prior models, low conditional diversity, and address it by modifying the base distribution. To quantify this aspect, they introduce a new metric, Average Feature Diversity.

**Questions:**

see above Strengths and Weaknesses.

**Ethical Concerns:**

["NO or VERY MINOR ethics concerns only"]

**Final Justification:**

I still affirm that this work fills some theoretical gaps. However, I am concerned that this writing style is overly exaggerated. Therefore, I maintain my rating unchanged.

**Limitations:**

yes.

**Quality:**

3

**Strengths And Weaknesses:**

Strengths

1.  This method performs well on conditional sampling tasks, generating high-quality outputs that stay aligned with the input. Its design helps capture important details between the source and target.

2. The method outperforms existing baselines across various conditional generation tasks. Also, there are enough ablation and experiments.

Weaknesses

The writing style of this paper is too overclaimed. The main contributions of this paper are the introduction of noise to the conditioning input to improve output diversity, and the proposal of a new metric to quantify conditional diversity. Other claims are mostly reformulations or combinations of existing ideas.

1. In Section 3, \emph{Have the bridge paths been fully explored?}, here the author an important question, whether the space of bridge paths has been fully explored, I was expected this work could provide some extension viewpoints, but the paper does not actually address it. The boundary conditions for alpha and beta are standard, and the decoupled parameterization is already known in stochastic interpolant literature. The authors merely test a specific gamma schedule and present limited ablations, falling short of a meaningful exploration, even no theoretical validation, why the new gamma schedule works?

2. Similar to Section 3, Section 4 \emph{Has the sampler space been fully explored?}, also poses a strong question—whether the sampler space has been fully explored—but offers a limited answer.

I personally find the questions posed in Section 3 and Section 4 are very interesting and important. However, in reading the paper, I feel that the proposed method only touches a small part of these spaces and does not fully engage with the broader implications of those questions. From my perspective, the work presents useful contributions, but they seem more modest than the framing suggests.
Overall, I believe the paper does introduce some new ideas. However, it falls significantly short of delivering on the promises implied by its title and section headings. The actual contributions are relatively minor compared to what is claimed.

In summary, while the technical contribution may merit acceptance, the overly exaggerated presentation makes me reluctant to recommend acceptance in its current form.

---

> ### Author Rebuttal · Authors · 2025-07-30
>
> We thank the reviewer for the detailed feedback. We are pleased to hear that you believe the paper introduces new ideas and makes useful contributions.
>
> Our claims are mostly combinations or reformulations of existing ideas. Surprisingly, though, there are gaps in the bridge literature that largely stem from the difficulty of combining ideas conceived under different conceptual frameworks (e.g. Stochastic interpolants[1] versus DDBM [2]). We don't believe it is too bold to ask "Have the bridge paths been fully explored?", to which the answer is a simple "no". However, we don't want to give the impression that our paper provides the final grand unified theory of bridge paths, and we will clarify the language accordingly by focusing and narrowing the scope of our statements in several places. Nevertheless, there are significant gaps in the existing literature that our work helps to fill.
>
> > The authors merely test a specific gamma schedule and present limited ablations, falling short of a meaningful exploration, even no theoretical validation, why the new gamma schedule works?
>
> We provided additional design guidelines in the Appendix D. The rationale and ablation studies are as follows:
>
> $\alpha_t$ and $\beta_t$. Theoretically, $\alpha_t$ and $\beta_t$ can be freely designed, and future work may explore alternative design choices. However, in this paper, we focus on the simple case where $\alpha_t = 1 - t$ and $\beta_t = t$. The rationale is as follows: consider the scenario where $\alpha_t = 1 - \beta_t$, which represents an interpolation along the line segment between $x_0$ and $x_1$. For the path
>
> $p_t^{(1)}(x) = \mathcal{N}((1 - \beta_t)x_0 + \beta_t x_1, \gamma_t^2 \mathbb{I})$, where $\beta_t$ is invertible, it is straightforward to construct another path
>
> $p_t^{(2)}(x) = \mathcal{N}((1 - t)x_0 + t x_1, \gamma_{\beta_t^{-1}}^2 \mathbb{I})$, which achieves the same objective function but uses a different distribution of $t$ during training. Based on this equivalence, setting $\alpha_t = 1 - t$ and $\beta_t = t$ is a reasonable choice.
>
> The shape of $\gamma_t$. We also conducted an ablation study on $\gamma_t$ with different shapes. Specifically, we assumed $\gamma_t$ has the form $\gamma_t=2 \gamma_{\max} \sqrt{t^k (1 - t^k)}$, the results are shown in Fig. (7). The results indicate that the best performance is achieved when k = 1, which is the exact setting used in this paper.
>
> $\gamma_{\max}$. Our ablation studies on $\gamma_{\max}$ demonstrate that the optimal values of $\gamma_{\max}$ are approximately $0.125$ or $0.25$. Furthermore, the sampling paths corresponding to different choices of $\gamma_t$ are shown in Fig. 8. Adding an appropriate amount of noise to the transition kernel helps in constructing finer details. This setting achieves the best performance across 3 different datasets: Edges 2 handbags, DIODE and ImageNet, demonstrating its generalization.
>
> $\epsilon_t$. We use the setting $\epsilon_t = \eta \left( \gamma_t \dot{\gamma}_t - \frac{\dot{\alpha}_t}{\alpha_t}\gamma_t^2 \right).$ The ablation studies on $\epsilon_t$ demonstrate that the optimal choice of $\eta$ for the DDBM-VP model is approximately $0.3$, while the best choice for the ECSI model with a Linear Path is around $1.0$. Additionally, we present sample paths and generated images under different $\eta$ settings to illustrate heuristic parameter tuning techniques. The results are shown in Figures 10, 11, and 12. Too small a value of $\eta$ results in the loss of high-frequency information, while too large a value of $\eta$ produces over-sharpened and potentially noisy sampled images.
>
> In our experiments, across different tasks, the superior performance is achieved by setting:
>
>  $$ p_{t|0,T}(x_t | x_0, x_T) = \mathcal{N}\left(x_t; (1-t)x_0 + t x_T, \frac{1}{4}\gamma_{\max}^2 t(1-t)\mathbb{I}\right) $$
>
> The best results are achieved by setting $\gamma_{\max}=0.125$ and $\gamma_{\max}=0.25$, and the best sampler is achieved by setting $\epsilon_t = \dot{\gamma_t} \gamma_t - \frac{\dot{\alpha_t}}{\alpha_t}\gamma_t^2=\gamma_{\max}^2 / 8$. This setting can be adapted to different kinds of image-to-image translation problems, e.g., edges to handbags, depth image to RGB image, deblurring.
>
> [1] Albergo M S, Boffi N M, Vanden-Eijnden E. Stochastic interpolants: A unifying framework for flows and diffusions[J]. arXiv preprint arXiv:2303.08797, 2023.
>
> [2] Zhou L, Lou A, Khanna S, et al. Denoising diffusion bridge models[J]. arXiv preprint arXiv:2309.16948, 2023.

---

> ### Comment · Reviewer_CniD · 2025-08-04
>
> I still affirm that this work fills some theoretical gaps. However, I am concerned that this writing style is overly exaggerated. Therefore, I maintain my rating unchanged.

---

### Official Review · Reviewer_NpZo · 2025-07-01

**Clarity:** 3
**Significance:** 2
**Originality:** 2
**Rating:** 4
**Confidence:** 3

**Summary:**

The paper proposes Endpoint-Conditioned Stochastic Interpolants (ECSI), an extension of Stochastic Interpolants to the endpoint-conditioned setting. The conditional dynamics of a stochastic process starting at $T=1$ and transforming a given sample ​into a sample from an unknown distribution at $T=0$ are derived while preserving the same marginals as a spatially linear stochastic interpolant. Additionally, a loss function is proposed to approximate these dynamics, enabling conditional sampling from the unknown distribution via SDE or ODE dynamics. To improve the diversity of generated samples, the paper proposes modifying the distribution at $T=1$ using a Gaussian convolution and introduces the Average Feature Distance (AFD), which sums Inception-V3 feature distances, to measure conditional diversity. Lastly, ECSI is evaluated empirically on Edges2handbags ( $64 \times 64$ ). DIODE-Outdoor datasets ( $256 \times 256$ ), deblurring on ImageNet ( $256 \times 256$ ) where ECSI is compared to diffusion bridge models and flow based methods.

**Questions:**

- The authors write in the abstract that “the proliferation of techniques based on incompatible mathematical assumptions have impeded progress.” What are these assumptions, and why do they impede the progress of diffusion bridge models for image-to-image translation?
- Could the authors clarify the advantages and conceptual differences of ECSI compared to the spatially linear interpolants presented in Albergo et al. [Section 4, 1]?
- How does ECSI perform compared to the unconditional setting of SI?
- What are the advantages of the proposed Average Feature Distance (AFD) compared to the Vendi Score (VS) [5] and other established metrics (see Weaknesses above)?
- Does the quantitative evaluation of ECSI using the Vendi Score (VS) support the authors’ claim that modifying the distribution at $T=1$ via a Gaussian convolution yields more diverse samples?
- What is the computational cost of calculating the Average Feature Distance (AFD)?

[5] Friedman et al. The Vendi Score: A Diversity Evaluation Metric for Machine Learning. Transactions on Machine Learning Research, 2023.

### Minor comments:
- Is it a typo that the authors write $\bar{f},\bar{g}$ in eq. (1) instead of $f,g$? The reviewer would assume with this notation $\bar{f}(t):=f(T-t)$
- There is a reference missing for the derivation of the reverse model in eq. (3) and eq. (4)
- There is a reference missing for the derivation of the denoising bridge score matching objective in eq. (5)
- $\log p_{t|0,T}$ is missing it's input in eq. (5)
- (line 105): $tx_1$ instead of $tx_0$
- (line 113): "by the SDE" instead of "by SDE"
- There is no reference in the main paper to the proof of Proposition 4.1 provided in the Appendix.
- (line 118): "we found that we could define [...] controlled the stochasticity" should be "we find that we can define [...] controls the stochasticity"

**Ethical Concerns:**

["NO or VERY MINOR ethics concerns only"]

**Final Justification:**

The reviewer assigns a score of 4 rather than 5, as it remains unclear why the authors do not compare their Endpoint-Conditioned Stochastic Interpolants framework to the Stochastic Interpolants framework without endpoint conditioning. Nonetheless, even without this comparison, the reviewer considers the work a valuable contribution that advances the theory of diffusion bridge models and is supported by strong empirical results.

**Limitations:**

yes

**Paper Formatting Concerns:**

No paper formatting concerns.

**Quality:**

3

**Strengths And Weaknesses:**

### Strengths
- To the best of the reviewer’s knowledge, the theoretical contribution presented in Proposition 4.1 constitutes an extension of the IS framework of Albergo et al. [1] to the endpoint-conditioned setting, and is therefore a valuable contribution in its own right.
- The paper tackles the important issue of diversity in conditional image generation and proposes a simple approach to improve diversity by modifying the base distribution.
- The empirical results suggest that the proposed ECSI method yields superior performance compared to Denoising Diffusion Bridge Models and Diffusion Bridge Implicit Models.

### Weaknesses
- The presented method, which extends the IS framework to the endpoint-conditioned setting, is not compared to the unconditional Stochastic Interpolants (IS) framework of Albergo et al. [1], neither at a conceptual level nor in terms of performance. An unconditional version of Proposition 4.1 has already been derived in [1], as [Corollary 2.18, 1] gives

\begin{equation}
dX_t = b(t,X_t)dt + \sqrt{2\epsilon}d W_t, \quad X_T \sim \pi_T
\end{equation}

and [eq. (4.4), 1] yields

\begin{equation}
b(t,x) = \dot{\alpha}(t)\mathbb{E}[X_0|X_t=x] + \dot{\beta(t)}\mathbb{E}[X_1|X_t=x] + (\gamma(t)+\frac{\epsilon(t)}{\gamma(t)})\mathbb{E}[Z|]X_t=x].
\end{equation}

The reviewer acknowledges that Proposition 4.1 is stated in the conditional setting $X_1=x_1$, and, in the reviewer’s opinion, it is not straightforward to derive Proposition 4.1 from Albergo et al. [1]. Moreover, the training objective proposed in this work differs from the loss functions presented in Albergo et al. [Eq. (4.6), 1], and hence, to the best of the reviewer’s knowledge, ECSI constitutes a novel method. However, the strong connection of Proposition 4.1 to the above derivations should be explicitly acknowledged, and a detailed discussion of the conceptual differences between ECSI and SI should be provided. Most importantly, the paper lacks a performance comparison between ECSI and the unconditional SI framework as described in Albergo et al. [1]. This comparison is especially relevant since Albergo et al. [1] already provides, for linear interpolants given by

\begin{equation}
X_t = \alpha(t)X_0 + \beta(t)X_1 + \gamma(t)z
\end{equation}

a detailed discussion of the choice of $\alpha(t),\beta(t)$ and $\gamma(t)$ in [Section 4, 1].

- The proposed metric, Average Feature Distance (AFD), to assess diversity in conditional image generation, is only compared to the Fréchet Inception Distance (FID) [2] and is justified solely by stating that assessing diversity of generated data via FID alone is insufficient. While the reviewer agrees with this point, it is a well-known issue, and several works have already addressed it. The paper lacks a detailed discussion of the differences and advantages of AFD compared to more recent methods for diversity quantification in generative models, such as Improved Recall [3], $\beta$-Recall [4], Vendi Score (VS) [5] and Feature Likelihood Divergence [6]. The reviewer suggests that the authors provide a thorough discussion of why AFD is preferable to the metrics proposed in [3,4,5,6] and/or include these metrics in their evaluation to support the claim that ECSI produces more diverse data. In particular, VS [5] is explicitly designed to measure the diversity of generated data and should be considered in both the conceptual discussion and the quantitative evaluation.

[1] Albergo et al. Stochastic interpolants: A unifying framework for flows and diffusions. arXiv preprint arXiv:2303.08797, 2023.

[2] Heuse et al. GANs Trained by a Two Time-Scale Update Rule Converge to a Local Nash Equilibrium. NIPS 2017.

[3] Kynkäänniemi et al. Improved Precision and Recall Metric for Assessing Generative Models. NeurIPS 2019.

[4] Alaa et al. How Faithful is your Synthetic Data? Sample-level Metrics for Evaluating and Auditing Generative Models. ICML 2022.

[5] Friedman et al. The Vendi Score: A Diversity Evaluation Metric for Machine Learning. Transactions on Machine Learning Research, 2023.

[6] Jiralerspong et al. Feature Likelihood Divergence: Evaluating the Generalization of Generative Models Using Samples. NeurIPS 2023.

---

> ### Author Rebuttal · Authors · 2025-07-30
>
> **Response to Weakness 1 and Question 2, 3**
>
> We sincerely thank the reviewer for their insightful comments and for recognizing the novelty of ECSI compared to Albergo et al. [1]. We acknowledge the strong connection between Proposition 4.1 in our work and the derivations for linear interpolants in Albergo et al. [1, Section 4]. We will revise the text to more explicitly emphasize the similarities as well as the conceptual differences.
>
> > However, the strong connection of Proposition 4.1 to the above derivations should be explicitly acknowledged, and a detailed discussion of the conceptual differences between ECSI and SI should be provided. Most importantly, the paper lacks a performance comparison between ECSI and the unconditional SI framework as described in Albergo et al. [1]. This comparison is especially relevant since Albergo et al. [1] already provides, for linear interpolants given by $X_t = \alpha(t) X_0 + \beta(t) X_1 + \gamma(t) Z$. a detailed discussion of the choice of $\alpha(t)$, $\beta(t)$, $\gamma(t)$ in [Section 4, 1].
>
> > Could the authors clarify the advantages and conceptual differences of ECSI compared to the spatially linear interpolants presented in Albergo et al. [Section 4, 1]?
>
> In short, both SI and ECSI can be seen as special cases of Conditioned SI. A key advantage of ECSI is its efficiency: while SI need to estimate two terms: $\mathbb{E}[x_0 \mid x_t]$ and $\mathbb{E}[x_1 \mid x_t]$, ECSI only estimates $\mathbb{E}[x_0 \mid x_t, y]$ (where $y=x_1$).
>
> First, SI can be extended to a conditional version. Conditioned Stochastic Interpolants build a marginal probability path $p_{t \mid y}$ using a mixture of interpolating densities: $p_{t \mid y}(x) = \int p_t(x_t \mid x_0, x_1) \pi(x_0, x_1 \mid y) dx_0 dx_1$, where $\pi(x_0,x_1 \mid y)$ is a joint distribution with marginals $\pi_{0 \mid y}(x_0 \mid y)$ and $\pi_{1 \mid y} (x_1 \mid y)$. For linear interpolants given by: $X_t=\alpha_t X_0 + \beta_t X_1 + \gamma_t z$. The conditional kernel $p_t(x_t \mid x_0, x_1)$ is  given by a Gaussian distribution: $p_t(x_t \mid x_0, x_1) = \mathcal{N}(\alpha_t x_0 + \beta_t x_1, \sigma_t^2 I), \forall ~ t \in [0, 1]$. Then we can sample from the conditional distribution $p_{0 \mid y}(x_0 \mid y)$ by running a stochastic process $p_{t \mid y} (x_t \mid y)$ from time $t = 1$ to $t = 0$, which is given by the following SDE:
>
> $$dx = b(t, x, y)dt + \sqrt{2 \epsilon_t} dW_t, \quad x_1 \sim p_{1 \mid y},$$
>
> where the drift term $b(t, x, y)$ is:
>
> $$b(t, x, y) = \dot{\alpha}_t \mathbb{E}[x_0 \mid x, y] + \dot{\beta}_t  \mathbb{E}[x_1 \mid x, y] +( \dot{\gamma}_t + \frac{\epsilon_t}{\gamma_t}) \mathbb{E}[z \mid x, y]$$
>
> As $y$ represents null conditioning, we recover the original sampler of Stochastic Interpolants. In the drift term, $\mathbb{E}[x_0 \mid x, y]$, $\mathbb{E}[x_1 \mid x, y]$ and $\mathbb{E}[z \mid x, y]$ are unknown, but we only need to estimate two of them, since $$\mathbb{E}[x_t \mid x_t, y]=\alpha_t \mathbb{E}[x_0 \mid x_t, y] + \beta_t \mathbb{E}[x_1 \mid x_t, y] + \gamma_t \mathbb{E}[z\mid x_t, y]=x_t$$
>
> We can further reduce the number of unknown term by endpoint-conditioning. Here as we replace condition $y$ to be endpoint $x_1$, the term $\mathbb{E}[x_1 \mid x, x_1]=x_1$. So we have:
>
> $$b(t, x, y)=\dot{\alpha}_t \mathbb{E}[x_0 \mid x, x_1] + \dot{\beta}_t x_1 + (\dot{\gamma}_t + \frac{\epsilon_t}{\gamma_t}) \mathbb{E}[z \mid x, x_1]$$
>
> This is exactly the sampler for ECSI in Proposition 4.1. Therefore, both SI and ECSI can be seen as special cases of Conditioned SI. A key advantage of ECSI is its efficiency: while SI need to estimate two terms: $\mathbb{E}[x_0 \mid x_t]$ and $\mathbb{E}[x_1 \mid x_t]$, ECSI only estimates $\mathbb{E}[x_0 \mid x_t, x_1]$.
>
>
> **Response to Weakness 2 and Question 4, 5**
>
> We thank the reviewer for this insightful feedback on the evaluation of diversity in diffusion models, including some works we were not aware of. Below, we discuss the applicability of each method. We find that Vendi score is a good fit for our evaluation, and we will add this metric to our diversity evaluation results.
>
>
> (1) $\beta$-Recall, Improved-Recall and Feature Likelihood Divergence are not applicable. In our work, we focus on the *conditional diversity* of images given a certain condition. For example, given a particular edge image, can we generate diverse images with different colors and textures. Therefore, we want to calculate the diversity of the generated set of images without knowing any prior distribution. Therefore, $\beta$-Recall, Improved-Recall and Feature Likelihood Divergence may not be applicable in our case. While we totally agree that the Vendi Score is well-suited in our case.
>
>
> (2) A comparison between AFD and other VS. Both AFD and the VS quantify diversity in the feature space of images, using features extracted from the Inception-V3 model. AFD measures the average pairwise Euclidean distance between feature vectors, making it sensitive to outliers. In contrast, the Vendi Score evaluates diversity by computing the effective number of unique feature patterns, based on the eigenvalues of the similarity matrix, emphasizing the overall structural diversity of the feature set. These metrics are complementary, capturing different aspects of diversity. For example, consider two sets of feature vectors: set $a=[2, 1, 1, 1]$, set $b=[1, 1.2, 1.3, 1.6]$, For AFD, the average pairwise Euclidean distance in set $a$ is larger due to the outlier $2$, resulting in $AFD(a)>AFD(b)$. For the Vendi Score, set $b$ has greater diversity due to its more varied feature values, so $VS(a)<VS(b)$.
>
>
> (3) Additional experiments. We have included an evaluation on different denoisers and samplers using AFD and VS, as shown in the table below. The results demonstrate that ECSI, particularly with the modified base density, consistently outperforms the DDBM checkpoint with DBIM or ECSI samplers across both AFD and VS metrics at varying NFEs (5, 10, 20).
>
>
> |                                   |   AFD | AFD    | AFD    | VS    | VS     | VS     |
> | --------------------------------- | ----: | ------ | ------ | ----- | ------ | ------ |
> |                                   | NFE=5 | NFE=10 | NFE=20 | NFE=5 | NFE=10 | NFE=20 |
> | DDDBM checkpoint + DBIM sampler   |  5.63 | 5.20   | 5.84   | 1.16  | 1.23   | 1.26   |
> | A: DDBM checkpoint + ECSI sampler |  5.11 | 5.70   | 6.04   | 1.15  | 1.20   | 1.23   |
> | B: ECSI checkpoint + sampler      |  6.00 | 6.05   | 6.25   | 1.22  | 1.25   | 1.28   |
> | B + Modified base density         |  **8.53** | **9.35**   | **9.65**   | **1.48**  | **1.63**   | **1.69**   |
>
> **Response to Question 1**
> > The authors write in the abstract that “the proliferation of techniques based on incompatible mathematical assumptions have impeded progress.” What are these assumptions, and why do they impede the progress of diffusion bridge models for image-to-image translation?
>
>
> We will re-word this claim in a clearer way. Recent diffusion bridge models excel in image translation but suffer from restricted design flexibility and complicated hyperparameter tuning, whereas Stochastic Interpolants offer greater flexibility but lack essential refinements. We show that these complementary strengths can be unified by interpreting all existing methods within a single SI-based framework.
>
> **Response to Question 6**
>
> > What is the computational cost of calculating the Average Feature Distance (AFD)?
>
>
> The computational cost of calculating the Average Feature Distance (AFD) is $O(n^2)$, compared to VS $O(n^3)$. AFD is more time efficient, but VS is less sensitive to outliers, as described above.
>
> **Response to Minor comments**
>
> > Is it a typo that the authors write $\bar{f}$, $\bar{g}$ in eq. (1) instead of $f$, $g$? The reviewer would assume with this notation $\bar{f}(t) := f(T-t)$
>
> In the original DDBM paper, they use the same notation for both forward process and reverse process. But we agree that it's better to use different notations for better clarification. We will revise this typo in the revised paper.
>
> > There is a reference missing for the derivation of the reverse model in eq. (3) and eq. (4)
>
> > There is a reference missing for the derivation of the denoising bridge score matching objective in eq. (5)
>
> We will add appropriate references to the derivations of the reverse model in eqs. (3) and (4), and the denoising bridge score matching objective in eq. (5), citing the relevant source in the revised manuscript.
>
>
> > $\log p_{t \mid 0, T}$ is missing it's input in eq. (5)
>
> > line 105, $t x_1$ instead of $t x_0$
>
> > (line 113): "by the SDE" instead of "by SDE"
>
> Thanks for the reviewer. We will revise those typos or mistakes in the revised version.
>
> > There is no reference in the main paper to the proof of Proposition 4.1 provided in the Appendix.
>
> We will include a reference in the main paper to direct readers to the proof of Proposition 4.1 in the Appendix.
>
>
>
>
>
>
>
> [1] Albergo et al. Stochastic interpolants: A unifying framework for flows and diffusions. arXiv preprint arXiv:2303.08797, 2023.
>
>
>
> [2] Heuse et al. GANs Trained by a Two Time-Scale Update Rule Converge to a Local Nash Equilibrium. NIPS 2017.
>
>
>
> [3] Kynkäänniemi et al. Improved Precision and Recall Metric for Assessing Generative Models. NeurIPS 2019.
>
>
>
> [4] Alaa et al. How Faithful is your Synthetic Data? Sample-level Metrics for Evaluating and Auditing Generative Models. ICML 2022.
>
>
>
> [5] Friedman et al. The Vendi Score: A Diversity Evaluation Metric for Machine Learning. Transactions on Machine Learning Research, 2023.
>
>
>
> [6] Jiralerspong et al. Feature Likelihood Divergence: Evaluating the Generalization of Generative Models Using Samples. NeurIPS 2023.

---

> > ### Comment · Reviewer_NpZo · 2025-08-01
> >
> > The reviewer appreciates the authors detailed response and clarifications, which address all but one concern. The reviewer remains of the opinion that a comparison between ECSI and SI would significantly strengthen the paper and still does not fully understand why this comparison was not considered in the first place. However, the reviewer acknowledges that a full comparison against SI may not be feasible within the limited time frame of the rebuttal period. Thank you for the effort to include an additional evaluation with respect to the VS. After considering the other reviews and the authors responses, I have decided to raise my score.

---

### Official Review · Reviewer_Bkd1 · 2025-07-02

**Clarity:** 3
**Significance:** 3
**Originality:** 3
**Rating:** 4
**Confidence:** 3

**Summary:**

This paper presents Endpoint-Conditioned Stochastic Interpolants (ECSI), a novel and unified framework that significantly expands the design space of diffusion bridge models for image-to-image (I2I) translation tasks. The authors systematically investigate the role of stochasticity across three key components: the transition kernel, the sampling SDE, and the base distribution. By extending stochastic interpolants with endpoint conditioning, preconditioning, and a more flexible sampling algorithm, the proposed method achieves new state-of-the-art results. A key contribution is the identification of the low conditional diversity problem in one-to-many translation tasks, which the authors address by modifying the base distribution and quantify using a newly proposed metric, Average Feature Distance (AFD). Experimental results on tasks like edges-to-handbags, depth-to-RGB, and deblurring demonstrate that ECSI surpasses previous methods in terms of image quality, sampling efficiency, and conditional output diversity.

**Questions:**

Please see the weakness.

**Ethical Concerns:**

["NO or VERY MINOR ethics concerns only"]

**Final Justification:**

The paper unifies a range of previous diffusion bridge models and introduces effective techniques for improving distribution matching. I believe the work has merit, and I will therefore keep my score at "Borderline accept."

**Limitations:**

Please see the weakness.

**Paper Formatting Concerns:**

Missing reference in line 487.

**Quality:**

3

**Strengths And Weaknesses:**

Strengths (Pros)

**Unified Framework and Expanded Design Space**: The paper presents a compelling and well-motivated framework (ECSI) that successfully unifies and expands upon previous diffusion bridge models like DDBM, DBIM, and Stochastic Interpolants. By systematically decoupling and exploring stochasticity in the transition kernel, the sampling process, and the base distribution, the authors provide a much broader and more flexible design space for I2I translation tasks. This is a strong conceptual contribution to the field.

**Significant and Comprehensive Empirical Gains**: The proposed method achieves impressive empirical results, establishing new state-of-the-art performance on multiple benchmark datasets. The improvements are not limited to a single dimension but are demonstrated across three critical axes:

*Sampling Quality and Speed*: The ECSI model and its sampler show a remarkable improvement in efficiency, achieving superior image quality (FID) with significantly fewer function evaluations (NFE) than prior art.

*Conditional Diversity*: A major strength is the identification and resolution of the often-overlooked issue of low conditional diversity. The proposed modification of the base distribution, coupled with the new Average Feature Distance (AFD) metric, provides a novel and effective way to generate more varied and realistic outputs for one-to-many translation tasks.

*Clarity and Rigor*: The paper is well-written and clearly structured. The theoretical connections between ECSI and prior models are well-explained, and the experimental section is thorough and convincing.


Weaknesses (Cons)

**Heuristic Design Principles and Lack of Theoretical Guidance**: While the paper successfully expands the design space, the exploration within this new space remains largely empirical and heuristic. The paper provides limited theoretical guidance or practical principles on how to optimally choose the bridge path parameters (αt, βt, γt) or the sampler noise schedule (ϵt) for a new, unseen task. This suggests that applying the framework to different problems might still require extensive and costly hyperparameter tuning, which somewhat tempers the practical benefits of the expanded flexibility.

**Limited Demonstration of Scalability**: The experiments, while thorough, are conducted on relatively small-scale datasets and resolutions (e.g., 64x64 handbags, 256x256 DIODE). It remains an open question whether the demonstrated gains in efficiency and quality will hold when the framework is scaled up to higher-resolution images or more complex, large-scale industrial applications. Demonstrating the method's effectiveness on a more challenging, large-scale benchmark would significantly strengthen the paper's claims about its general applicability.

**Insufficient Analysis of Interacting Stochasticity Sources**: The paper introduces two key sources of randomness: the path stochasticity (controlled by γt, which determines the noise z_t in the forward process) and the sampler stochasticity (controlled by ϵt during reverse sampling). However, the relationship and potential interplay between these two noise sources are not deeply analyzed. For instance, could a wider, more stochastic path (larger γt) be compensated for by a more deterministic sampler (smaller ϵt), or vice-versa? A more in-depth analysis or ablation study on how these two forms of stochasticity interact would provide deeper insights into the model's behavior and could lead to even better design principles.

---

> ### Author Rebuttal · Authors · 2025-07-30
>
> **Response to Weakness 1**
>
> We provided additional design guidelines in the Appendix D. The rationale and ablation studies are as follows:
>
> $\alpha_t$ and $\beta_t$. Theoretically, $\alpha_t$ and $\beta_t$ can be freely designed, and future work may explore alternative design choices. However, in this paper, we focus on the simple case where $\alpha_t = 1 - t$ and $\beta_t = t$. The rationale is as follows: consider the scenario where $\alpha_t = 1 - \beta_t$, which represents an interpolation along the line segment between $x_0$ and $x_1$. For the path
>
> $p_t^{(1)}(x) = \mathcal{N}((1 - \beta_t)x_0 + \beta_t x_1, \gamma_t^2 \mathbb{I})$, where $\beta_t$ is invertible, it is straightforward to construct another path
>
> $p_t^{(2)}(x) = \mathcal{N}((1 - t)x_0 + t x_1, \gamma_{\beta_t^{-1}}^2 \mathbb{I})$, which achieves the same objective function but uses a different distribution of $t$ during training. Based on this equivalence, setting $\alpha_t = 1 - t$ and $\beta_t = t$ is a reasonable choice.
>
> The shape of $\gamma_t$. We also conducted an ablation study on $\gamma_t$ with different shapes. Specifically, we assumed $\gamma_t$ has the form $\gamma_t=2 \gamma_{\max} \sqrt{t^k (1 - t^k)}$, the results are shown in Fig. (7). The results indicate that the best performance is achieved when k = 1, which is the exact setting used in this paper.
>
> $\gamma_{\max}$. Our ablation studies on $\gamma_{\max}$ demonstrate that the optimal values of $\gamma_{\max}$ are approximately $0.125$ or $0.25$. Furthermore, the sampling paths corresponding to different choices of $\gamma_t$ are shown in Fig. 8. Adding an appropriate amount of noise to the transition kernel helps in constructing finer details. This setting achieves the best performance across 3 different datasets: Edges 2 handbags, DIODE and ImageNet, demonstrating its generalization.
>
> $\epsilon_t$. We use the setting $\epsilon_t = \eta \left( \gamma_t \dot{\gamma}_t - \frac{\dot{\alpha}_t}{\alpha_t}\gamma_t^2 \right).$ The ablation studies on $\epsilon_t$ demonstrate that the optimal choice of $\eta$ for the DDBM-VP model is approximately $0.3$, while the best choice for the ECSI model with a Linear Path is around $1.0$. Additionally, we present sample paths and generated images under different $\eta$ settings to illustrate heuristic parameter tuning techniques. The results are shown in Figures 10, 11, and 12. Too small a value of $\eta$ results in the loss of high-frequency information, while too large a value of $\eta$ produces over-sharpened and potentially noisy sampled images.
>
> In our experiments, across different tasks, the superior performance is achieved by setting:
>
>  $$ p_{t|0,T}(x_t | x_0, x_T) = \mathcal{N}\left(x_t; (1-t)x_0 + t x_T, \frac{1}{4}\gamma_{\max}^2 t(1-t)\mathbb{I}\right) $$
>
> The best results are achieved by setting $\gamma_{\max}=0.125$ and $\gamma_{\max}=0.25$, and the best sampler is achieved by setting $\epsilon_t = \dot{\gamma_t} \gamma_t - \frac{\dot{\alpha_t}}{\alpha_t}\gamma_t^2=\gamma_{\max}^2 / 8$. This setting can be adapted to different kinds of image-to-image translation problems, e.g., edges to handbags, depth image to RGB image, deblurring.
>
> **Response to Weakness 2**
>
> We appreciate the reviewer’s feedback regarding the scalability of ECSI. Our experiments were conducted on three datasets—64x64 Handbags, 256x256 DIODE, and 256x256 ImageNet—which are standard benchmarks widely used in prior works, including I2SB [1], DDBM [2], and DBIM [3]. These datasets were chosen to ensure fair comparisons with existing methods. While we acknowledge the importance of evaluating scalability on higher-resolution or more complex industrial applications, the consistent performance improvements observed across these datasets provide strong evidence of ECSI’s efficiency and quality. To further address scalability concerns, we are open to extending our evaluation to larger-scale, higher-resolution datasets in future work and can include a discussion of this in the paper to highlight potential applicability in such settings.
>
> **Response to Weakness 3**
>
> We thank the reviewer for highlighting the need for deeper analysis of the interaction between path stochasticity (controlled by $\gamma_t$) and sampler stochasticity (controlled by $\epsilon_t$). In our experiments, we explicitly address this interplay by selecting the sampler stochasticity based on the path stochasticity, as defined by the relation $\epsilon_t= \eta(\gamma_t \dot{\gamma}_t - \frac{\dot{\alpha}_t}{\alpha_t} \gamma_t^2)$, where $\eta$ controls the stochasticity strength at inference time. This is also an innovation of ECSI and this selection has not been demonstrated in previous work. We observed that for different noise levels of path stochasticity (controlled by $\gamma_t$), the best $\eta$ is achived at $\eta=1$. We will include these details in the main text and emphasize our contribution on specifically designing of $\epsilon_t$.
>
>
> [1] Liu G H, Vahdat A, Huang D A, et al. I $^ 2$ SB: Image-to-Image Schr\" odinger Bridge[J]. arXiv preprint arXiv:2302.05872, 2023.
>
> [2] Zhou L, Lou A, Khanna S, et al. Denoising diffusion bridge models[J]. arXiv preprint arXiv:2309.16948, 2023.
>
> [3] Zheng K, He G, Chen J, et al. Diffusion bridge implicit models[J]. arXiv preprint arXiv:2405.15885, 2024.

---

> > ### Comment · Reviewer_Bkd1 · 2025-08-05
> >
> > Thanks for the rebuttal; it resolved most of my concerns. I look forward to seeing the evaluation extended to larger-scale, higher-resolution datasets in future work. I will keep my score at "borderline accept."

---

### Official Review · Reviewer_WLqt · 2025-07-02

**Clarity:** 2
**Significance:** 1
**Originality:** 1
**Rating:** 4
**Confidence:** 3

**Summary:**

This paper presents a unified approach to expand the design space of diffusion bridge models by incorporating preconditioning, endpoint conditioning, and an optimized sampling algorithm into Stochastic Interpolants (SIs). The proposed framework advances the state of image-to-image translation by improving both image quality and sampling efficiency across diverse tasks. Furthermore, it addresses the overlooked issue of low sample diversity under fixed conditions through a quantitative analysis of output diversity, demonstrating how base distribution modifications can enhance results.

**Questions:**

1. What is the difference between the training scheme of the ECSI to the previous bridge models?
2. Will the proposed ECSI sampler mitigate the sampling discretization error?
3. Does the marginal distribution of the ECSI sampler coincide with the underlying true marginal distribution from a rigorous Fokker-Planck equation analysis?

**Ethical Concerns:**

["NO or VERY MINOR ethics concerns only"]

**Final Justification:**

After reading all the review comments, I found the proposed bridge model is unified approach and novel enough.

**Limitations:**

Clarification of the paper is limited.

**Quality:**

2

**Strengths And Weaknesses:**

**Strengths**

1. The proposed bridge model is unified approach and surpasses pervious models (e.g. DDBM, DDIM) in handling varying  characteristics.
1. The experimental results are highly illustrative, providing clear empirical support.

**Weaknesses**

1. The novelty of ECSI primarily resides in introducing additional noise along the sampling path compared to the original bridge models' sampling process. While this design enhances diversity, the underlying intuition is not entirely novel, as similar concepts of noise augmentation in diffusion sampling have been discussed in prior work (e.g., EDM).
2. It remains unclear whether the training scheme of ECSI differs substantially from that of DDBM. If there are differences, the approach resembles applying noised stochastic sampling to DDBM models, analogous to endpoint conditioning techniques that utilize $x_0$-prediction for original DDBM frameworks.
3. The paper lacks theoretical analysis on the convergence behavior of the newly proposed method, which is a notable gap in the methodological justification.

---

> ### Author Rebuttal · Authors · 2025-07-30
>
> **Response to Weakness 1**
>
> Thank you for your feedback. We agree that noise augmentation in diffusion sampling, as explored in works like EDM and Stochastic Interpolants, shares similarities with our approach. However, a key aspect of ECSI is ensuring the marginal distribution remains consistent by satisfying the Fokker-Planck equation through our derived SDEs, which we believe is a significant contribution.
>
> Additionally, we argue that ECSI’s novelty extends beyond noise scheduling. By modifying the base distribution, we introduce a new strategy to enhance diversity, which is not sufficiently addressed by adding noise alone, as shown in our experiments. This approach is distinct and, to our knowledge, novel, with results demonstrating its effectiveness in improving diversity for diffusion bridge models.
>
> **Response to Weakness 2**
>
> We appreciate your insights and the opportunity to clarify the distinctions between ECSI and DDBM. While ECSI shares some conceptual similarities with DDBM, its training scheme diverges in key ways, particularly through the integration of noised stochastic sampling and a novel modification of the base distribution. This modification enhances diversity beyond what endpoint conditioning or x0-prediction achieves in the original DDBM framework.
>
> We view ECSI as an advancement akin to an "EDM" version of DDBM, with additional innovations. Specifically: (1) ECSI unifies multiple diffusion bridge models (I2SB, DDBM, DBIM, and EDM), as demonstrated in Table 8; (2) it decouples parameters in the transition kernel, enabling a broader design space; and (3) it supports a more flexible noise schedule during inference. Besides,  we specifically designed a noise scheduling at inference time, i.e., $\epsilon_t= \eta(\gamma_t \dot{\gamma}_t - \frac{\dot{\alpha}_t}{\alpha_t} \gamma_t^2)$, which achieves better results in the experiments.
>
> Furthermore, We also proposed a novel strategy that modifies the base distribution. Intuitively, this strategy interpolates between standard bridge models and unconditional diffusion models. We found that this strategy diversifies the sampling path in a complementary manner to other stochasticity controls. This strategy significantly improved the sampler diversity.
>
> **Response to Weakness 3**
>
> In our paper, we provide a rigorous proof for Proposition 1, which establishes the theoretical foundation of ECSI and demonstrates that I2SB [1], DDBM [2], DBIM [3], and EDM can be viewed as special cases within our unified framework, see Appendix B. Due to space constraints, detailed proofs are provided in Appendix B. Please let us know if you had a different type of convergence result in mind that we failed to address in the Appendix.
>
> **Response to Question 1**
>
> We adopt the same $x_0$ prediction and preconditioning as DDBM [2]. The training details could be found in Appendix F.
>
> **Response to Question 2**
>
> We acknowledge the importance of minimizing discretization errors in diffusion models. However, like other samplers for continuous diffusion and diffusion bridge models, the ECSI sampler does not fully eliminate sampling discretization error. To our knowledge, no existing sampler for these models can completely mitigate this issue. To reduce discretization error, we recommend increasing the number of sampling steps or adopting higher-order discretization schemes, as these approaches have proven effective in practice.
>
>
> **Response to Question 3**
>
> Thank you for your question regarding the marginal distribution of the ECSI sampler. We confirm that the marginal distribution of the ECSI sampler coincides with the true underlying marginal distribution, as established through a rigorous Fokker-Planck equation analysis. Specifically, Lemma B.2 in Appendix B provides a detailed proof demonstrating how the inclusion of a divergence-free term ensures that the marginal distribution remains consistent with the true distribution. We are happy to provide further clarification or expand on this analysis if needed.
>
> [1] Liu G H, Vahdat A, Huang D A, et al. I $^ 2$ SB: Image-to-Image Schr\" odinger Bridge[J]. arXiv preprint arXiv:2302.05872, 2023.
>
> [2] Zhou L, Lou A, Khanna S, et al. Denoising diffusion bridge models[J]. arXiv preprint arXiv:2309.16948, 2023.
>
> [3] Zheng K, He G, Chen J, et al. Diffusion bridge implicit models[J]. arXiv preprint arXiv:2405.15885, 2024.
>
> [4] Karras T, Aittala M, Aila T, et al. Elucidating the design space of diffusion-based generative models[J]. Advances in neural information processing systems, 2022, 35: 26565-26577.

---

> > ### Comment · Reviewer_WLqt · 2025-08-05
> > **Thanks for the rebuttal**
> >
> > I thank the authors for their detailed rebuttal addressing my concerns. After going thought all the review comments and the authors' rebuttal, I am convinced that ECSI is novel and differs from DDBM in its training scheme. Consequently, I am willing to raise my rating.

---

### Comment · Area_Chair_VhPr · 2025-08-04

Dear reviewers,

Please post your response as soon as possible, if you haven’t done it, to allow time for discussion with the authors. All reviewers should respond to the authors' rebuttal to confirm it has been read.

Thanks,

AC

---

### Note · Authors · 2025-08-13

We thank the reviewers for their constructive feedback. The main points raised concerned comparisons with Stochastic Interpolants, diversity metrics, ablation studies and writing style.

- **Comparison with Stochastic Interpolants**: Reviewer NpZo suggest that a comparison with Stochastic Interpolants would significantly strengthen the paper. In the rebuttal, we added a conceptual comparison between Endpoint-conditioned Stochastic Interpolants (ECSI) and Stochastic Interpolants (SI), highlighting the advantages of ECSI over SI. A more detailed discussion will be included in the revised manuscript.

- **Diversity metrics**: We compared our proposed metric, Average Feature Distance (AFD), with the Vendi Score (VS), suggested by the reviewer NpZo. Additional evaluations using VS are provided, further supporting our claim that modifying the base distribution yields more diverse samples.

- **Ablation studies**: We provided additional design guidelines and ablation studies in the Appendix D. These studies examine the impact of varying $\alpha_t$, $\beta_t$, the shape of $\gamma_t$, and $\epsilon_t$, offering practical guidelines for optimizing hyperparameters.

- **Writing style**: We will refine the language in the revised version by narrowing the scope of certain statements and ensuring clarity.

Overall, the reviewers recognized that our work proposes a novel and extended framework that integrates complementary ideas and addresses important theoretical gaps. The proposed framework advances the state of image-to-image translation by improving both image quality and sampling efficiency across diverse tasks. Moreover, ECSI draws attention to the overlooked issue of low sample diversity under fixed conditions and effectively addresses it through modifying base distribution.

---

### Decision · Program_Chairs · 2025-09-17

**Decision:**

Accept (poster)

**Comment:**

This paper presents a unified framework, ECSI, for diffusion bridge models, aiming to improve image-to-image translation. The key idea is to expand the design space by preconditioning, endpoint conditioning, and a flexible sampling algorithm. The main strengths are the unification of prior models and strong empirical results showing better image quality, sampling speed, and output diversity. A primary weakness is that some design principles are heuristic. However, the paper's solid performance and its novelty to improving sample diversity are convincing reasons for acceptance.

During the rebuttal, reviewers raised several points. Key issues included the method's novelty compared to prior work (WLqt), the lack of comparison to the unconditional stochastic interpolants framework (NpZo), and an overclaimed writing style (CniD). The authors effectively addressed most concerns. They clarified the theoretical novelty and differences in the training scheme, convincing Reviewer WLqt. For Reviewer NpZo, they provided a conceptual comparison to stochastic interpolants. They also added new experiments using the Vendi score to validate their diversity claims. After reading all discussions, I agree with the reviewers and recommend accepting the paper, as the authors thoroughly addressed the reviewers’ concerns.